# Language Model Augmented Semi-Supervised Statistical Inference

**Xinrui Ruan** [1]  **Yingfei Wang** [2]  **Waverly Wei** [3]  **Jingshen Wang** [* 1]

## Abstract

Semi-supervised statistical inference plays a key role in biomedical research, where labeled data often have higher quality but are limited due to costly clinical annotation. Yet, existing semi-supervised statistical inference methods rely heavily on structured variables and strictly matched covariates between labeled and unlabeled datasets–limitations ill-suited for the heterogeneity and unstructured nature of real-world biomedical data. Modern biomedical studies increasingly collect unstructured data (clinical notes, patient audio and video recordings), with inconsistent protocols across datasets causing covariate misalignment (for instance, detailed medication histories may be recorded in one study but not another). Recent advances in pre-trained multimodal large language models (LLMs), which excel at handling unstructured data, present an attractive potential solution. To transform this potential into rigorous semi-supervised statistical inference methods for biomedical research, two key challenges must be addressed: (1) How can we reliably integrate LLMs to enhance semi-supervised inference efficiency without compromising statistical validity? (2) How can those efficiency gains persist despite mismatched covariates between labeled and unlabeled datasets? In this paper, we tackle these challenges by systematically *calibrating pseudo-labels* provided LLMs with a novel *prediction-invariance identification* strategy. Our resulting semi-supervised inference framework improves parameter estimation efficiency while maintaining full statistical validity, as demonstrated through our theoretical results and illustrated in a case study for identifying key

biomarkers in Alzheimer's disease detection with speech data.

## 1. Introduction

### 1.1. Motivation and our contributions

In biomedical research, semi-supervised statistical inference (SSI) has emerged as an important extension of semi-supervised learning (SSL), shifting the focus from outcome prediction to uncovering meaningful associations between outcomes and key variables of interest using both labeled and unlabeled data. This shift is partly due to the fact that while SSI typically aims to improve prediction accuracy by leveraging abundant unlabeled data alongside limited, costly expert-labeled clinical data (Van der Vaart, 2000; Beaulieu-Jones et al., 2016), biomedical researchers are more concerned with estimating these associations and rigorously quantifying their uncertainty. These associations can reveal, for instance, factors linked to elevated disease risk, offering insights into underlying biological mechanisms and informing clinical decision-making. SSI methods are thus designed to meet this need by improving the efficiency and precision of association estimation–such as producing narrower confidence intervals and increasing the power to detect meaningful relationships–through statistically principled use of unlabeled data (Zhang et al., 2019; Azriel et al., 2022).

Despite recent progress, the application of SSI in modern biomedical research is limited by two practical challenges commonly encountered in real-world biomedical datasets. **First**, existing SSI frameworks are tailored for analyzing structured data, such as demographic variables, lab measurements, and diagnosis codes, whereas many real world biomedical data have unstructured modalities, such as free-text clinical notes (Shankar et al., 2025), acoustic patterns in speech (Fraser et al., 2015), and temporal dynamics in medical videos (Ahrens et al., 2013). **Second**, existing SSI frameworks often assume strict covariate alignment between labeled and unlabeled datasets, an assumption that is rarely met in practice. This is because biomedical research studies are often conducted under heterogeneous protocols, resulting in systematic covariate mismatches across collected datasets (Day & Khoshgoftaar, 2017). For example,

[1]Division of Biostatistics, University of California, Berkeley [2]Department of Information Systems and Operations Management, Foster School of Business, University of Washington [3]Department of Data Sciences and Operations, University of Southern California. Correspondence to: Jingshen Wang <jingshenwang@berkeley.edu>.

*Proceedings of the 43rd International Conference on Machine Learning*, Seoul, South Korea. PMLR 306, 2026. Copyright 2026 by the author(s).

datasets from different hospital systems may include similar outcome labels (e.g., depression diagnosis), yet differ substantially in how unstructured inputs are collected. One prominent case is the MIMIC-III database from Beth Israel Deaconess Medical Center, which contains rich narrative clinical notes written by psychiatrists and other providers (Johnson et al., 2016). In contrast, data sources such as the National Ambulatory Medical Care Survey (NAMCS) often rely on structured checkboxes and brief text fields completed during outpatient visits (Centers for Disease Control and Prevention, 2024). Such discrepancies in data representation introduce challenges for SSI across datasets.

This paper addresses these two challenges by leveraging recent advances in large-scale pre-trained multimodal large language models (LLMs), which have demonstrated impressive capabilities in processing, understanding, and predicting outcomes from heterogeneous and unstructured biomedical data. Notable examples include BioGPT (Luo et al., 2022) and PubMedBERT (Gu et al., 2021) for biomedical text, Med-PaLM (Singhal et al., 2025) and LLaVA-Med (Li et al., 2023) for multimodal clinical reasoning, and TabLLM (Hegselmann et al., 2023) for learning from tabular data in healthcare records. These models enable integration of diverse modalities while retaining strong generalization across domains.

However, naively applying LLMs to facilitate SSI poses several challenges: LLMs are primarily optimized for predictive accuracy rather than preserving statistical inference validity, and the predictions they produce may not align with the labeled outcomes. Furthermore, classical machine learning methods such as gradient boosting typically outperform transformer-based architectures on structured tabular data with well-aligned covariates (Shwartz-Ziv & Armon, 2022; Grinsztajn et al., 2022). In many biomedical settings, datasets may contain both structured and unstructured covariates. In such cases, relying solely on LLMs may fail to fully exploit predictive signals. As a result, how to integrate LLMs into semi-supervised inference frameworks in a statistically principled manner, ensuring consistent parameter estimation, valid uncertainty quantification, and improved statistical efficiency, remains an open question.

In this paper, we propose the **L**anguage model **A**ugmented **S**emi-supervised **S**tatistical inference (LASS) framework to address the statistical challenges outlined above. We organize our contributions into three key components, each targeting a specific challenge:

1. *Leveraging LLMs for unstructured data and calibrating predictions for valid and efficient inference.* In Section 3.1, we introduce a novel framework for integrating LLM predictions under matched covariates. Since raw LLM outputs may deviate from the true labels, we employ classical machine learning methods

to calibrate the predictions. We then construct pseudo-labels by combining the observed labels and calibrated predictions, and use these pseudo-labels to estimate the statistical model of interest.

2. *Discovering prediction-invariant regions to address covariate misalignment assisted by LLMs.* In Section 3.2, we extend our framework to settings with covariate misalignment arising from heterogeneous data collection protocols. We introduce an algorithm to discover *Prediction Invariance Regions* where the average LLM predictions remain stable across datasets. This allows us to selectively incorporate information from the mismatched covariates, enabling robust statistical inference.

3. *Provable efficiency gains while preserving statistical validity.* In Section 4, we establish theoretical guarantees ensuring that the integration of calibrated LLM predictions leads to valid inference. Theorems 4.2 and 4.4 provide rigorous asymptotic guarantees for the proposed estimators in Section 3. We also characterize the conditions under which the proposed estimators have an efficiency gain over the existing estimators in Corollary 4.3.

### 1.2. Related literature

*Semi-supervised inference.* When labeled data are limited, semi-supervised inference offers a principled way to leverage unlabeled data to improve the efficiency of parameter estimation. Recent methods aim to flexibly incorporate unlabeled data without relying on strong parametric assumptions. (Kawakita & Kanamori, 2013) proposed a weighted likelihood approach based on density ratio estimation. (Chakrabortty & Cai, 2018) introduced adaptive estimators that leverage auxiliary covariate information to reduce variance. These ideas have been extended to model performance evaluation (Gronsbell & Cai, 2018), general M-estimation frameworks (Song et al., 2024), and high-dimensional linear models (Tony Cai & Guo, 2020). More recently, (Angelopoulos et al., 2023a) introduced the Prediction-Powered Inference (PPI) framework, which imputes missing labels using black-box machine learning models and constructs valid confidence intervals. Follow-up work has improved PPI's efficiency through reweighting (Angelopoulos et al., 2023b) and extended it to cross-validated predictors (Zrnic & Candès, 2024b). Nevertheless, most existing methods are not designed to handle real-world biomedical data characterized by multimodal features and covariate misalignment.

*LLM-based prediction.* LLMs have demonstrated remarkable predictive performance across diverse tasks by leveraging massive pretraining and contextual understanding. Early applications focused on natural language tasks such as text

classification, question answering, and summarization using models like BERT and GPT-2. Recent advances, including GPT-3, PaLM, and LLaMA, have shown strong zero-shot and few-shot capabilities, making LLMs adaptable to new prediction tasks with minimal supervision (Brown et al., 2020; Chowdhery et al., 2023; Touvron et al., 2023). In scientific and medical domains, domain-adapted models such as BioGPT and Galactica incorporate specialized corpora to improve relevance and accuracy (Luo et al., 2022; Taylor et al., 2022). Beyond language, LLMs have been extended for reasoning over structured data (Hegselmann et al., 2023) and complex decision-making (Yao et al., 2023). However, most LLM-based prediction pipelines have limited emphasis on uncertainty quantification or statistical guarantees.

## 2. Problem setup

### 2.1. Statistical setup with notations

We now describe the data structure for Semi-supervised Statistical Inference (SSI) and the statistical estimation target with unstructured data and possible covariate misalignment. Assume we observe a sample of $N$ unlabeled subjects with covariate information $\{X_i, Z_i\}_{i=1}^N$, and a sample of $n$ labeled subjects with $\{Y_i, X_i, Z_i, U_i\}_{i=N+1}^{N+n}$. In the presence of misalignment, we consider the case that the labeled data are of higher quality and include additional covariates $U_i$ that are not available in the unlabeled data. Here, $Y_i$ denotes the true label, $X_i \in \mathcal{X} \subset \mathbb{R}^p$ is the covariate of primary interest, and $Z_i$ represents the auxiliary covariates. We let $U_i$ denotes additional covariates that are available only in the labeled dataset. Both $Z_i$ and $U_i$ may be high-dimensional or unstructured. When there is no covariate misalignment, such as when the labeled and unlabeled datasets are collected under the same standard protocol or when labels are missing completely at random, the labeled dataset can be written as $\{Y_i, X_i, Z_i\}_{i=N+1}^{N+n}$, without observing additional covariates $U$. Our proposed method applies to both settings with aligned covariates (Section 3.1) and misaligned covariates (Section 3.2).

Throughout the main text, we assume that $\{Y_i, X_i, Z_i, U_i\}_{i=1}^{N+n}$ are independently and identically distributed according to some distribution $\mathbb{P}$, with $Y_i$ and $U_i$ unobserved for the unlabeled subjects ($1 \leq i \leq N$). Extensions to settings with covariate shift between the labeled and unlabeled samples are discussed in the Appendix.

In biomedical studies, researchers and clinicians are often interested in quantifying the uncertainty in associations between key variables and outcomes, as these associations are crucial for understanding underlying biological mechanisms and informing clinical decision-making. For instance, in SSI with electronic health records, a small subset of pa-

tients may have expert-annotated cognitive scores (labels) alongside structured variables like age and medication use (primary covariates) and unstructured data such as clinical notes or speech recordings (auxiliary covariates). Estimating the associations between specific linguistic features in the unstructured data and cognitive outcomes, while rigorously *quantifying uncertainty despite limited labels*, can provide valuable insights into early signs of neurodegenerative disease and guide timely interventions.

To formalize the problem, let $X$ denote the primary covariate(s) of interest (the argument extends to any subset of $X$). Because the outcome $Y$ and the covariates can be on different scales, for instance, $Y$ may be binary, we relate $Y$ to a linear predictor in $X$ through a known link function $g(\cdot)$. Rather than imposing the full generalized linear model (GLM) assumption which restricting the conditional distribution of outcome variables (e.g., $\mathbb{E}[Y \mid X] = g(X^\top \beta)$), we define the target parameter $\beta^*$ as the solution to the following moment condition

$$\beta^* \ : \ \mathbb{E}[X(Y - g(X^\top \beta))] = 0. \tag{1}$$

This definition of $\beta^*$ avoids specifying the conditional distribution of $Y \mid X$, making the estimation more robust to model misspecification than assuming a full GLM. We also hope to note that $\beta^*$ is a commonly adopted parameter in the current SSI literature (Liu et al., 2023; Cai et al., 2024). To estimate $\beta^*$, one can solve the generalized estimating equation only using the labeled data, obtaining the an simple GLM estimator $\hat{\beta}_{\mathrm{GLM}}$ that satifies $\frac{1}{n} \sum_{i=N+1}^{N+n} X_i\{Y_i - g(X_i^\top \beta)\} = 0$. As this estimator completely ignores unlabeled data, it is statistically inefficient.

### 2.2. A concrete example: Extracting clinical insights from DementiaBank with SSI

To provide concrete practical guidance on the statistical setup in Section 2.1, DementiaBank is well-suited for illustrating our semi-supervised framework as it contains a small, deeply annotated cohort with AD diagnosis labels (e.g., AD, probable AD) and several much larger speech corpora from general dementia patients that lack these labels (Lanzi et al., 2023). The **labeled** Pitt corpus contains 452 participants with clinician-confirmed AD diagnoses ($Y_i$); demographic covariates such as race, sex, and education, together with NLP-derived linguistic features like lexical complexity, filler-word ratio, and utterance length ($X_i$); auxiliary unstructured information $Z_i$ in the form of full speech transcripts and audio recordings; and additional cognitive test recordings, including Hamilton test scores, conversational transcripts from the fluency, recall, and sentence construction tests ($U_i$) that are measured only in this labeled set. In contrast, **unlabeled** corpora drawn from other research sites contribute roughly 1,800 additional speech samples in total. These unlabeled records include the same

primary covariates $X_i$ and transcripts $Z_i$, yet they lack detailed dementia diagnoses $Y_i$ and the detailed cognitive tests $U_i$.

Estimating how strongly the demographic and linguistic markers $X_i$ predict different AD types is clinically important, because neurologists and speech-language pathologists can rely on evidence such as reduced lexical complexity or increased filler-word use to decide who should undergo costly and possibly invasive neuroimaging for definitive diagnosis (McKhann et al., 2011). Yet the labeled Pitt corpus alone is too small for precise estimates, while discarding the roughly 1,800 additional unlabeled speech samples would waste critical information. Our proposed SSI framework (LASS) thus leverages the abundant $X_i$ and $Z_i$ in the unlabeled datasets, while properly accounting for the absence of $Y_i$ and $U_i$, can reduce uncertainty around the $X_i$ to AD association, giving clinicians more precise, actionable insights for AD screening based on easily accessible speech data.

## 3. Method

In this section, we introduce our **L**anguage model **A**ugmented **S**emi-supervised **S**tatistical inference (LASS) framework. Section 3.1 describes the method under matched covariates, while Section 3.2 extends the framework to the more general setting of covariate misalignment, where the labeled and unlabeled data structures differ due to heterogeneous data collection protocols.

### 3.1. LASS: Matched covariates when labels are missing completely at random

When the covariate structures between labeled and unlabeled data are matched, implying that experts may have randomly selected certain patient profiles for labeling, our proposed LASS framework begins by using a pre-trained LLM to generate predicted labels for all subjects based on their covariates. Specifically, given covariates $\{X_i, Z_i\}$, we query the LLM using a carefully designed prompt $\mathcal{P}$, and extract the predicted label $Y^*_{\text{xz},i}$ from the answer:

$$X_i, Z_i \xrightarrow[\text{zero-shot prediction}]{\text{LLM}} Y^*_{\text{xz},i}, \quad \text{for } 1 \le i \le n. \quad (2)$$

Since we are interested in the most likely prediction, rather than a creative generation, we set the LLM's temperature to $0$ to disable its generative randomness. The resulting predicted label reflects the pre-trained LLM's encoded knowledge about the association between the covariates and the outcome.

However, directly incorporating LLM-predicted labels does not necessarily achieve optimal statistical efficiency. This is because although LLM predictions may be informative, they

may not align well with the labeled outcomes, which can lead to inconsistency or even inflate the estimation variance.

To effectively harness the information encoded in LLM-predicted labels, we propose a calibration strategy aiming to understand the conditional correlation between LLM-predicted labels and the true labels. All four functions below are estimated using classical machine learning methods. Specifically, we first estimate two conditional mean functions: the true outcome model $m(x) = \mathbb{E}[Y \mid X = x]$, and the predicted outcome model $m^*_{\text{xz}}(x) = \mathbb{E}[Y^*_{\text{xz}} \mid X = x]$. Next, we estimate two key quantities used for calibration:

$$\gamma_{\text{xz}}(x) = \text{Cov}\left(Y^*_{\text{xz}}, Y \mid X = x\right),$$
$$\nu_{\text{xz}}(x) = \text{Var}\left(Y^*_{\text{xz}} \mid X = x\right),$$

where $\gamma_{\text{xz}}(x)$ represents the conditional covariance between the true labels and the predicted labels, and $\nu_{\text{xz}}(x)$ serves as a normalization factor. We then define the estimated "*optimal calibration weighting function*" as

$$\widehat{\omega}^*_{\text{xz}}(x) = \hat{\gamma}_{\text{xz}}(x)/\hat{\nu}_{\text{xz}}(x),$$

which is the ratio of these two estimated quantities. Intuitively, a large value of $|\omega^*_{\text{xz}}(x)|$ indicates that the LLM-predicted label is more informative about the true label given $X = x$. This suggests that the LLM effectively captures predictive information from the unstructured covariates $Z$, and a higher weight should be assigned. As we will show in Section 4, this weight also quantifies the potential efficiency gain that can be achieved by incorporating LLM-based predictions into the estimation procedure.

We then construct a pseudo label $\tilde{Y}_{\text{xz},i}$ for each subject $1 \le i \le N + n$ by:

$$\widetilde{Y}_{\text{xz},i} \equiv \begin{cases} \hat{f}_{\text{xz},i}, & \begin{array}{l} 1 \le i \le N \\ \text{(Unlabeled)} \end{array} \\ Y_i + \frac{N}{n}\left(Y_i - \hat{f}_{\text{xz},i}\right), & \begin{array}{l} N+1 \le i \le N+n \\ \text{(Labeled)} \end{array} \end{cases}$$
$$\tag{3}$$

where the imputed value $\hat{f}_{\text{xz},i}$ is defined as $\hat{f}_{\text{xz},i} = \hat{m}(X_i) + \hat{\omega}^*_{\text{xz}}(X_i)\left(Y^*_{\text{xz},i} - \hat{m}^*_{\text{xz}}(X_i)\right)$. We then obtain the estimator $\hat{\beta}_{\text{LASS},\text{xz}}$ of the target parameter $\beta$ by solving the moment equation:

$$\hat{\beta}_{\text{LASS},\text{xz}} : \frac{1}{N+n}\sum_{i=1}^{N+n} X_i\left(\tilde{Y}_{\text{xz},i} - g(X_i^\top \beta)\right) = 0. \tag{4}$$

The variance of $\hat{\beta}_{\text{LASS},\text{xz}}$ is estimated by the sandwich formula $\hat{\mathbb{V}}_{\text{LASS},\text{xz}} = \hat{A}^{-1}\hat{B}\hat{A}^{-1}$, where

$$\hat{A} = \frac{1}{N+n}\sum_{i=1}^{N+n} g'\left(X_i^\top \hat{\beta}_{\text{LASS},\text{xz}}\right) X_i X_i^\top,$$

$$\hat{B} = \frac{1}{N+n}\sum_{i=1}^{N+n}\left(\widetilde{Y}_{\text{xz},i} - g\left(X_i^\top \hat{\beta}_{\text{LASS},\text{xz}}\right)\right)^2 X_i X_i^\top.$$

**Algorithm 1.** LASS WITH MATCHED COVARIATES

1: **Label Prediction:** For each subject $1 \leq i \leq N+n$, predict label $Y^*_{\mathrm{xz},i}$ via a pre-trained LLM with input $(X_i, Z_i)$.
2: **LLM Prediction Calibration:**
3:    Fit $m(x) = \mathbb{E}[Y \mid X = x]$ and $\gamma_{\mathrm{xz}}(x) = \mathrm{Cov}(Y^*_{\mathrm{xz}}, Y \mid X = x)$ using labeled data.
4:    Fit $m^*_{\mathrm{xz}}(x) = \mathbb{E}[Y^*_{\mathrm{xz}} \mid X = x]$ and $\nu_{\mathrm{xz}}(x) = \mathrm{Var}(Y^*_{\mathrm{xz}} \mid X = x)$ using all data.
5:    Compute $\widehat{\omega}^*_{\mathrm{xz}}(x) = \widehat{\gamma}_{\mathrm{xz}}(x)/\widehat{\nu}_{\mathrm{xz}}(x)$.
6: **Calibrated Label Construction:** Construct labels $\widetilde{Y}_{\mathrm{xz},i}$ using Eq. (3).
7: **Semi-supervised Inference:** Estimate $\hat{\beta}_{\mathrm{LASS},\mathrm{xz}}$ by solving Eq. (4), and construct confidence intervals using the variance estimator.

Finally, a $(1-\alpha)$ confidence interval for $\hat{\beta}_{\mathrm{LASS},\mathrm{xz}}$ is constructed by $\left[\hat{\beta}_{\mathrm{LASS},\mathrm{xz}} \pm Z_{1-\alpha/2}\sqrt{\hat{\mathbb{V}}_{\mathrm{LASS},\mathrm{xz}}}\right]$. We provide a pseudocode summary of the procedure in Algorithm 1.

### 3.2. LASS: Misaligned covariates when datasets are collected under heterogeneous protocols

In this section, we extend the LASS framework introduced in Section 3.1 to robustly incorporate additional covariates $U$ that are available only in the labeled dataset. Incorporating additional information into LLM-based predictions can be beneficial, as they may contain valuable information for label prediction. For example, in the DementiaBank study, integrating multiple types of chat transcripts, each emphasizing different aspects of speech, can help the LLM more comprehensively assess a patient's status. Analogous to Equation 2, we define $Y^*_{\mathrm{xzu},i}$ as the predicted label for labeled subject $i$ using $(X_i, Z_i, U_i)$ as input, for $N+1 \leq i \leq N+n$. However, due to the covariate misalignment between the labeled and unlabeled datasets, and the nonlinear nature of the LLM prediction map, directly replacing $Y^*_{\mathrm{xz},i}$ with $Y^*_{\mathrm{xzu},i}$ in the pseudo-label construction of Equation 3 for all $N+1 \leq i \leq N+n$ may lead to inconsistency of the estimator. To preserve consistency while leveraging the information in $U$, we propose a two-step robust procedure.

We begin by identifying a **Prediction Invariance Region** (**PIR**). While the existence of such a region is **not** a requirement for LASS, its identification enables LASS to more effectively extract information from LLM-based predictions compared to those outside this region. The PIR is defined as:

$$\mathbb{E}\left[Y^*_{\mathrm{xzu}} \mid X \in \mathcal{A}\right] = \mathbb{E}\left[Y^*_{\mathrm{xz}} \mid X \in \mathcal{A}\right].$$

This condition ensures that, within region $\mathcal{A}$, incorporating $U$ does not systematically shift the LLM's average prediction relative to using only $(X, Z)$. The region $\mathcal{A}$ is not uniquely defined, and our goal is to select a region with

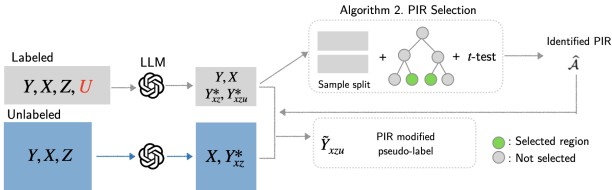

*Figure 1.* Illustration of Method in Section 3.2. We note that the properties of LASS do not need the existence of PIR.

**Algorithm 2.** PREDICTION INVARIANT REGION SELECTION

1: Compute the global statistic on labeled data: $T = (\bar{Y}^*_{\mathrm{xz}} - \bar{Y}^*_{\mathrm{xzu}})/(s/\sqrt{n})$.
2: If $|T| \leq t_{\alpha/2, n-1}$, set $\hat{\mathcal{A}} \leftarrow \mathcal{X}$ and **stop**.
3: Randomly split labeled indices $\{N+1, \ldots, N+n\}$ into two equal parts $\mathcal{I}_1, \mathcal{I}_2$.
4: Train a depth-$d$ CART on $\mathcal{I}_1$ to model $\mathbb{E}[Y^*_{\mathrm{xz}} - Y^*_{\mathrm{xzu}} \mid X = x]$. Let the leaves be $\ell_1, \ldots, \ell_K$, and initialize $\mathcal{M} \leftarrow \varnothing$.
5: For each leaf $k = 1, \ldots, K$, define $\mathcal{I}_{2,k} = \{i \in \mathcal{I}_2 : X_i \in \ell_k\}$ and compute $T_k = \frac{\bar{Y}^*_{\mathrm{xz}, \mathcal{I}_{2,k}} - \bar{Y}^*_{\mathrm{xzu}, \mathcal{I}_{2,k}}}{s_{\mathcal{I}_{2,k}}/\sqrt{|\mathcal{I}_{2,k}|}}$.
6: If $|T_k| \leq t_{\alpha/(2K), |\mathcal{I}_{2,k}|-1}$, add leaf $k$ to $\mathcal{M}$.
7: Return $\hat{\mathcal{A}} = \bigcup_{k \in \mathcal{M}} \ell_k$.

high probability mass $\mathbb{P}(X \in \mathcal{A})$, while ensuring fast convergence and low computational cost. Since the labeled and unlabeled data are drawn from the same population as described in Section 2.1, we can identify such a region by comparing the predicted labels on the labeled dataset $\{X_i, Y^*_{\mathrm{xz},i}, Y^*_{\mathrm{xzu},i}\}_{i=N+1}^{N+n}$.

To this end, we introduce the **PIR Selection** algorithm to identify $\hat{\mathcal{A}}$ detailed in Algorithm 2. Specifically, we begin by splitting the labeled dataset into two subsamples. In the first subsample, we apply CART to identify candidate regions, thereby reducing the computational burden of searching over the covariate space. In the second subsample, we use the CART-defined partition to divide the covariate space and conduct multiple $t$-tests to evaluate and select the prediction invariance regions. This sample-splitting strategy ensures that the region-level means in the second subsample are estimated without bias, following the "honest" principle of (Athey & Imbens, 2016).

Once $\hat{\mathcal{A}}$ is identified, we modify the procedure from Section 3.1 to accommodate the covariate misalignment setting. Specifically, we define the adjusted working functions as $m^*_{\mathrm{xzu}}(x) = \mathbb{E}[Y^*_{\mathrm{xzu}} \mid X = x]$, $\gamma_{\mathrm{xzu}}(x) = \mathbb{C}\mathrm{ov}(Y^*_{\mathrm{xzu}}, Y \mid X = x)$, $\nu_{\mathrm{xzu}}(x) = \frac{N}{N+n}\mathbb{V}\mathrm{ar}(Y^*_{\mathrm{xzu}} \mid X = x) + \frac{n}{N+n}\mathbb{V}\mathrm{ar}(Y^* \mid X = x)$, and $\omega^*_{\mathrm{xzu}}(x) = \gamma_{\mathrm{xzu}}(x)/\nu_{\mathrm{xzu}}(x)$. The original functions $m(x)$, $m^*_{\mathrm{xz}}(x)$ and $\omega^*(x)$ remain as in Section 3.1. We

then construct the adjusted pseudo-label as:

$$
\widetilde{Y}_{\text{xzu},i} = 
\begin{cases}
\hat{m}(X_i) + \hat{\omega}^*_{\text{xz}}(X_i)\left(Y^*_{\text{xz},i} - \hat{m}^*_{\text{xz}}(X_i)\right), \\
\qquad \text{if } 1 \le i \le N,\ X_i \notin \hat{\mathcal{A}}, \\
\hat{m}(X_i) + \hat{\omega}^*_{\text{xzu}}(X_i)\left(Y^*_{\text{xz},i} - \hat{m}^*_{\text{xz}}(X_i)\right), \\
\qquad \text{if } 1 \le i \le N,\ X_i \in \hat{\mathcal{A}}, \\
Y_i + \dfrac{N}{n}\left(Y_i - \hat{m}(X_i) - \hat{\omega}^*_{\text{xz}}(X_i)\left(Y^*_{\text{xz},i} - \hat{m}^*_{\text{xz}}(X_i)\right)\right), \\
\qquad \text{if } N+1 \le i \le N+n,\ X_i \notin \hat{\mathcal{A}}, \\
Y_i + \dfrac{N}{n}\left(Y_i - \hat{m}(X_i) - \hat{\omega}^*_{\text{xzu}}(X_i)\left(Y^*_{\text{xzu},i} - \hat{m}^*_{\text{xz}}(X_i)\right)\right), \\
\qquad \text{if } N+1 \le i \le N+n,\ X_i \in \hat{\mathcal{A}}.
\end{cases}
\tag{5}
$$

Compared to Equation (3), the pseudo-label $\widetilde{Y}_{\text{xzu},i}$ modifies the construction for individuals with $X_i \in \hat{\mathcal{A}}$ to incorporate extra information from $U$ in a robust manner, while preserving consistency. In the next step, we replace $\widetilde{Y}_{\text{xz},i}$ with $\widetilde{Y}_{\text{xzu},i}$ in the estimating equation (4) to obtain $\hat{\beta}_{\text{LASS},\text{xzu}}$, followed by statistical inference described previously. The procedure of Section 3.2 is in Figure 1.

## 4. Theoretical investigation

In this section, we investigate the theoretical properties of the proposed estimators introduced in Section 3. Theorems 4.2 and 4.4 demonstrate the consistency and asymptotic normality of $\hat{\beta}_{\text{LASS},\text{xz}}$ from Section 3.1 and $\hat{\beta}_{\text{LASS},\text{xzu}}$ from Section 3.2, respectively. In Corollary 4.3, we compare the asymptotic variance of $\hat{\beta}_{\text{LASS},\text{xz}}$ with that of the naive GLM estimator, as well as existing semi-supervised (Chakrabortty & Cai, 2018) and prediction-powered (Angelopoulos et al., 2023b) approaches. We begin by outlining the necessary regularity conditions.

**Assumption 4.1. (Regularity conditions)**

1. **Asymptotic regime.** The sample sizes $n$ and $N$ satisfy $\frac{n}{N+n} \to \pi \in (0,1]$ as $N+n \to \infty$.

2. **Bounded moments.** $\mathbb{E}\left[\|X\|^4\right] < \infty$ and $\mathbb{E}\left[Y^4\right] < \infty$.

3. **Language-model prediction.** The pretrained language model returns a bounded numeric prediction $Y^*_W$ for each input $W$. Predictions are generated separately for different subjects.

4. **Consistency of nuisance function estimators.** The estimated nuisance functions satisfy $\|\hat{m} - m\|_{L_2(\mathbb{P}_{\mathcal{X}})} \xrightarrow{p} 0$, $\|\hat{m}^* - m^*\|_{L_2(\mathbb{P}_{\mathcal{X}})} \xrightarrow{p} 0$, and $\|\hat{\omega}^* - \omega^*\|_{L_2(\mathbb{P}_{\mathcal{X}})} \xrightarrow{p} 0$.

Assumption 4.1.3 ensures that the predicted label is a well-defined i.i.d. random variable, while Assumption 4.1.4 can be satisfied by many machine learning methods under mild conditions, including $k$-nearest neighbors, kernel regression, and random forests (Scornet et al., 2015). We provide a more detailed discussion of Assumption 4.1.4 in Appendix C.2.

**Theorem 4.2** (Consistency and asymptotic normality of $\hat{\beta}_{\text{LASS},\text{xz}}$). *Under Assumptions 4.1 and the standard regularity conditions for estimation equations in Chapter 5 of (Van der Vaart, 2000), we have $\hat{\beta}_{\text{LASS},\text{xz}} \xrightarrow{p} \beta^*$, and:*

$$
\sqrt{n}(\hat{\beta}_{\text{LASS},\text{xz}} - \beta^*) \xrightarrow{d} N(0, \mathbb{V}_{\text{LASS},\text{xz}}),
$$

*where $\mathbb{V}_{\text{LASS},\text{xz}} = A^{-1}BA^{-1}$, and:*

$$
A = \mathbb{E}\left[g'\left(X^\top \beta^*\right) XX^\top\right],
$$

$$
B = \underbrace{\mathbb{E}\left[\frac{1}{\pi}\left(Y - g\left(X^\top \beta^*\right)\right)^2 XX^\top\right]}_{\text{Variance of simple GLM estimator}}
$$

$$
- \underbrace{\mathbb{E}\left[\left(\frac{1}{\pi} - 1\right)\left(m(X) - g\left(X^\top \beta^*\right)\right)^2 XX^\top\right]}_{\text{Efficiency gain from machine learning imputation}}
$$

$$
- \underbrace{\mathbb{E}\left[\left(\frac{1}{\pi} - 1\right)\frac{\mathbb{C}ov^2(Y^*_{\text{xz}}, Y \mid X)}{\mathbb{V}ar(Y^*_{\text{xz}} \mid X)} XX^\top\right]}_{\text{Efficiency gain from LLM prediction with calibration}}.
$$

Theorem 4.2 formally establishes the asymptotic properties of the estimator in Section 3.1 and decomposes its asymptotic variance into three terms, highlighting two sources of efficiency gain over the simple GLM estimator. In Theorem 4.2, the asymptotic variance depends on $\pi$ through the factors $1/\pi$ and $1/(\pi - 1)$, indicating that the labeled-to-unlabeled sample ratio affects the estimator's first-order variance. Intuitively, the labeled sample anchors valid inference, while the unlabeled sample contributes efficiency gains through calibrated pseudo-labels. As the unlabeled sample becomes relatively larger, the contribution of the unlabeled-data gain terms increases, allowing informative and well-calibrated LLM predictions to produce greater variance reduction. We formally compare its efficiency with existing estimators in the next corollary.

**Corollary 4.3** (Efficiency Comparison). *Define $\mathbb{V}_{\text{GLM}}$ as the asymptotic variance of a standard GLM fit using labeled data only, $\mathbb{V}_{\text{SS}}$ as the asymptotic variance under conventional SSI with a nonparametric imputation $\hat{m}(x)$, and $\mathbb{V}_{\text{PPI++}}$ (Angelopoulos et al., 2023b) as the asymptotic variance under the PPI++ estimator with a single global power parameter using the same prediction $Y^*_{\text{xz}}$. Under the conditions of Theorem 4.2, we have*

$$
\mathbb{V}_{\text{LASS},\text{xz},(j,j)} \le \mathbb{V}_{\text{SS},(j,j)} \le \mathbb{V}_{\text{GLM},(j,j)},
$$

$$
\mathbb{V}_{\text{LASS},\text{xz},(j,j)} \le \mathbb{V}_{\text{PPI++},(j,j)},
$$

*for each $1 \le j \le p$. Moreover, $\mathbb{V}_{\text{LASS},\text{xz},(j,j)} < \mathbb{V}_{\text{SS},(j,j)}$ if $\mathbb{C}ov(Y^*_{\text{xz}}, Y \mid X = x) \ne 0$ for some $x$, and $\mathbb{V}_{\text{LASS},\text{xz},(j,j)} < \mathbb{V}_{\text{PPI++},(j,j)}$ if $\mathbb{V}ar\{\omega^*_{\text{xz}}(X)\} > 0$.*

Corollary 4.3 shows that the proposed estimator $\hat{\beta}_{\text{LASS},\text{xz}}$ is asymptotically no worse than the existing benchmarks.

Strict efficiency improvement requires additional conditions. For conventional SSI, the calibrated LLM prediction must provide predictive information beyond the covariates used by SSI, as captured by $\mathbb{C}\mathrm{ov}(Y^*_{\mathrm{xz}}, Y \mid X = x) \neq 0$. For PPI++, strict improvement requires that the optimal calibration of the LLM-predicted label $\omega^*_{\mathrm{xz}}(x)$ is nonconstant in $x$. We defer the comparison with estimation based on deep-learning (DL) pseudo-labels to Appendix G. We now turn to the asymptotic properties of $\hat{\beta}_{\mathrm{LASS}, \mathrm{xzu}}$, introduced in Section 3.2, under the covariate misalignment setting.

**Theorem 4.4** (Consistency and asymptotic normality of $\hat{\beta}_{\mathrm{LASS}, \mathrm{xzu}}$). *Assume that $Y^*_{xzu}$ satisfies Assumption 4.1.2 and that $\hat{m}^*_{xzu}$ satisfies Assumption 4.1.4. Let $\mathcal{A}$ denote the limiting region of $\hat{\mathcal{A}}$. Then, under the conditions of Theorem 4.2 and the additional regularity conditions stated in Appendix E, we have $\hat{\beta}_{LASS, xzu} \xrightarrow{p} \beta^*$ and*

$$\sqrt{n}(\hat{\beta}_{LASS, xzu} - \beta^*) \xrightarrow{d} N(0, \mathbb{V}_{LASS, xzu}),$$

*where $\mathbb{V}_{LASS, xzu} = A^{-1}_{xzu} B_{xzu} A^{-1}_{xzu}$, and*

$$A_{xzu} = \mathbb{E}\left[ g'\left( X^\top \beta^* \right) X X^\top \right],$$

$$B_{xzu} = \mathbb{E}\left[ \frac{1}{\pi} \left( Y - g\left( X^\top \beta^* \right) \right)^2 X X^\top \right]$$

$$- \mathbb{E}\left[ \left( \frac{1}{\pi} - 1 \right) \left( m(X) - g\left( X^\top \beta^* \right) \right)^2 X X^\top \right]$$

$$- \mathbb{E}\left[ \left( \frac{1}{\pi} - 1 \right) \frac{\mathbb{C}\mathrm{ov}^2(Y^*_{xz}, Y \mid X)}{\mathbb{V}\mathrm{ar}(Y^*_{xz} \mid X)} X X^\top \Big| X \notin \mathcal{A} \right] \mathbb{P}(X \notin \mathcal{A})$$

$$- \mathbb{E}\Big[ \left( \frac{1}{\pi} - 1 \right) \frac{\mathbb{C}\mathrm{ov}^2(Y^*_{xzu}, Y \mid X)}{(1-\pi)\mathbb{V}\mathrm{ar}(Y^*_{xzu} \mid X) + \pi\mathbb{V}\mathrm{ar}(Y^*_{xz} \mid X)}$$

$$X X^\top \Big| X \in \mathcal{A} \Big] \cdot \mathbb{P}(X \in \mathcal{A}).$$

*Moreover, if $\gamma_{xz}(x)\omega^*_{xz}(x) < \gamma_{xzu}(x)\omega^*_{xzu}(x)$ for $x \in \mathcal{A}$, then for each $1 \leq j \leq p$, we have $\mathbb{V}_{LASS, xzu,(j,j)} < \mathbb{V}_{LASS, xz,(j,j)}$.*

Theorem 4.4 demonstrates that $\hat{\beta}_{\mathrm{LASS}, \mathrm{xzu}}$ is more efficient than $\hat{\beta}_{\mathrm{LASS}, \mathrm{xz}}$, provided that the prediction $Y^*_{\mathrm{xzu}}$, which incorporates additional covariates $U$, is more informative than $Y^*_{\mathrm{xz}}$ for the label $Y$. In our empirical studies, we verify that the set $\mathcal{A}$ is non-empty and that there exists some $x$ for which $\mathbb{C}\mathrm{ov}(Y^*_{\mathrm{xz}}, Y \mid X = x) > 0$.

# 5. Empirical studies

In this section, we evaluate the performance of LASS using synthetic data with matched covariates (Section 5.3), synthetic data with mismatched covariates (Section 5.4), and a real-world case study (Section 5.5) based on Alzheimer's disease speech from DementiaBank and self-extracted YouTube data.

**Summary of our findings**. **First**, LASS achieves higher estimation efficiency with valid coverage than three benchmarks, a labeled-only method, a conventional SSI method

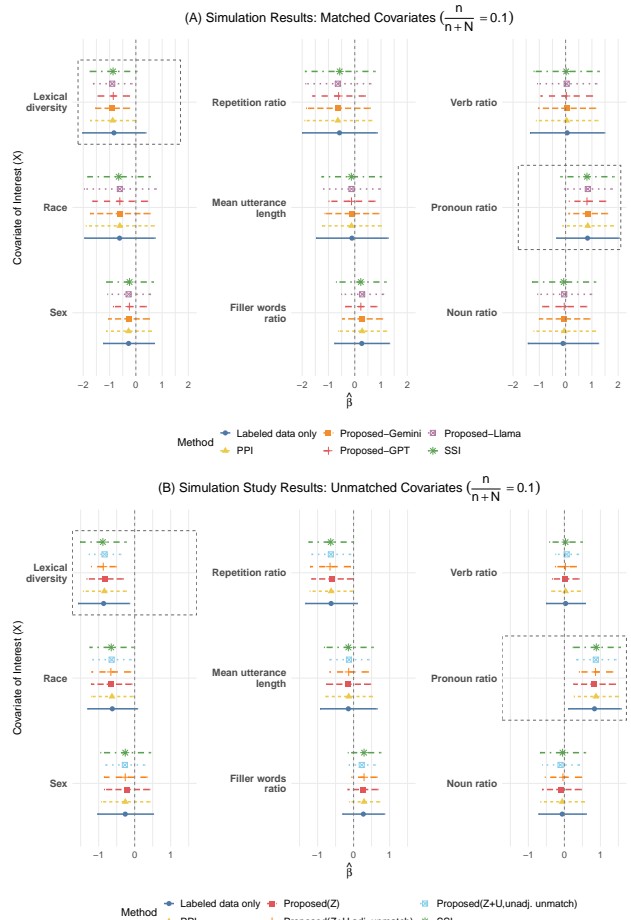

*Figure 2.* Comparison of various methods with matched covariates (Panel A) and unmatched covariates (Panel B).

without unstructured covariates, and prediction-powered inference (PPI; Angelopoulos et al., 2023a), across simulation and case study settings (Figures 2–5). **Second**, under covariate mismatch, identifying the prediction invariance region further improves estimation efficiency and yields accurate point estimates (Figure 5). **Third**, unlike benchmarks that cannot effectively leverage unlabeled data, LASS robustly identifies linguistic features associated with Alzheimer's disease, demonstrating practical utility for early screening and clinical decision-making using DementiaBank and real-world YouTube AD patient speech data (Figure 5).

## 5.1. Labeled and unlabeled data extraction

**Study background and motivation**. Linguistic biomarkers from speech provide a promising, non-invasive approach for early Alzheimer's disease diagnosis (Fraser et al., 2015; Eyigoz et al., 2020; Kourtis et al., 2019), but available AD speech data are limited by privacy concerns and dependence on invasive imaging for accurate labeling (MacWhinney, 2014; Luz et al., 2021; Martínez-Nicolás et al., 2021; Jack et al., 2010). We therefore leverage unstructured, pub-

| | Notation | Outcome $Y$ | Covariate of interest $X$ | Auxiliary covariates | |
|---|---|---|---|---|---|
| | | | | $Z$ | $U$ |
| **Pitt data** | Labeled | ✓ | ✓ | ✓ | ✓ |
| | | diagnosis (1=AD, 0=no AD) | Sex, race NLP features | chat transcripts | Hamilton rating,other transcripts |
| **Hopkins data** | Unlabeled | ✗ | ✓ | ✓ | ✗ |
| | | NA | Sex,race NLP features | chat transcripts | NA |
| **Self-extracted Youtube data** | Unlabeled | ✗ | ✓ | ✓ | ✗ |
| | | NA | Sex, race NLP features | chat transcripts | NA |

*Table 1.* Data structure for our empirical application.

licly available speech transcripts from DementiaBank and YouTube (MacWhinney, 2014; Yang et al., 2022; Tóth et al., 2015) and apply our LASS method to identify linguistic features predictive of AD, demonstrating practical utility for early screening and clinical decision-making.

**Labeled data from DementiaBank Pitt study**. DementiaBank includes the Pitt study (Becker et al., 1994), which consists of audio recordings from subjects completing cognitive tests, explicitly labeled with Alzheimer's disease diagnoses. This data contains structured covariates (e.g., demographic variables, linguistic features) and unstructured covariates (conversational transcripts). **Unlabeled data from DementiaBank Hopkins study and YouTube**. DementiaBank also includes the Hopkins study (Sebastian et al., 2018), which contains audio recordings of patients with dementia undergoing cognitive tests, though without AD diagnosis. Due to the limited sample size of the audio recordings stored in DementiaBank, we further extracted unlabeled data from YouTube, as it provides publicly accessible multimodal content, including videos, audio and textual transcripts of AD individuals.

**Brief YouTube data extraction pipeline**. To extract YouTube video data that is related to AD, we start by utilizing the YouTube Data API v3 to systematically assemble a comprehensive multimodal dataset capturing interactions involving AD and dementia patients. We queried YouTube videos using a two-tier keyword strategy, primary AD terms combined with context-specific interaction keywords, to ensure broad coverage of Alzheimer's-related content across diverse settings. Each video was then analyzed with Gemini models using a structured prompt to systematically extract rich, unstructured qualitative information about dementia patient interactions. We defer the details of the data extraction pipeline and the components of the structured prompt to Appendix Section A.2. **Structured linguistic feature extraction**. To extract structured linguistic features, we employ natural language processing (NLP) techniques to derive quantitative linguistic features from each transcript. The details are in Appendix A.2.

## 5.2. Study design for simulation studies

To assess performance under known ground truth, we generate synthetic labeled and unlabeled populations (5,000 each) using a zero-shot approach that preserves the statistical and linguistic properties of the original data, and then sample from these populations. We run 100 simulation iterations for both matched and mismatched covariate settings, with total sample sizes $N + n \in \{200, 600, 1000\}$ (fixing $n = 100$) and covariates including demographics and NLP-derived linguistic features.

**Evaluation metrics**. To compare efficiency, we evaluate confidence band widths, where narrower bands indicate higher precision. We report the average width of 95% confidence intervals across covariates for each method, along with ATE estimates averaged over simulation iterations.

**Benchmarks**. For both matched and unmatched covariates settings, we consider three benchmarks: logistic regression using only labeled data, classical SSI that imputes labels via machine learning, and recent prediction-powered inference (PPI) that also uses AI-based prediction to improve SSI efficiency (Angelopoulos et al., 2023a).

**Various prompt design and different foundation models**. To evaluate the performance of our methods under different prompting techniques, we compare three prompt engineering methods to generate the predicted outcomes using LLMs. The simulation results are summarized in Appendix Figure 3. As the predicted outcomes can vary depending on the choice of LLM, we evaluate the performance of our proposed method using predicted outcomes generated by three distinct models: (1) GPT-3.5-turbo, (2) Gemini-2.5-flash, and (3) LLaMA-3.1-8B.

## 5.3. Simulation results for matched covariates

We first evaluate LASS in the matched-covariate setting described in Section 3.1, using only covariates available in both datasets, including unstructured conversational transcripts from the "Cookie Theft" test that capture linguistic markers of cognitive impairment (Cummings, 2019). Figure 2 (A) shows that our proposed method yields uniformly shorter confidence intervals, indicating improved efficiency, with the GPT-based variant achieving the largest gains and

identifying lexical complexity and pronoun ratio as significant AD risk factors. Appendix Figure 6 (A) further illustrates these gains by showing strata-specific calibration weights that reflect correlations between LLM-generated and labeled outcomes and enable selective information borrowing.

### 5.4. Simulation results for unmatched covariates

We next consider settings with unmatched auxiliary covariates, where the Pitt dataset contains unique structured (Hamilton scores) and unstructured (fluency, recall, and sentence construction transcripts) features. For $N + n = 1000$, Figure 2(B) shows that LASS, which explicitly accounts for covariate mismatch, yields consistent estimates with narrower confidence intervals than benchmarks by selectively borrowing information from the prediction-invariant region. **Additional results.** We include additional benchmarks for PPI and its LLM-augmented variant (PPI+LLM). The results are summarized in Appendix Table 2. An additional comparison of LASS using CART versus random forests is presented in Appendix Table 6. Furthermore, to minimize the effect of hallucination, we use structured prompts that explicitly constrain the LLM output format. We summarize the results in Appendix Table 4.

### 5.5. Real world data case studies

We evaluate our method on two real-world case studies: DementiaBank combined with self-extracted YouTube data, and the IMDb Large Movie Review dataset. Focusing on unmatched covariates, we report results in Appendix Figure 5 and illustrate the prediction-invariant region (PIR) in Appendix Figure 6 (B). Our analysis identifies lexical complexity as a marker of cognitive decline, an effect missed by benchmarks, consistent with linguistic simplification in AD, and we further demonstrate the generalizability of our framework on IMDb (details in Appendix Section B).

## 6. Discussion

We acknowledge that the current empirical evaluation does not cover the full range of biomedical modalities, including images, videos, longitudinal measurements, and multicenter data sources. An important direction for future work is to evaluate LASS in broader biomedical and multimodal settings, such as neuroimaging, multimodal Alzheimer's disease datasets, and radiology imaging datasets, especially those collected across multiple centers. Such evaluations would help assess the transportability and robustness of LASS and clarify when calibrated pseudo-labels from foundation models can improve statistical efficiency while preserving valid inference in complex biomedical studies.

## Impact Statement

This paper presents work whose goal is to advance the field of Machine Learning. There are many potential societal consequences of our work, none which we feel must be specifically highlighted here.

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

# A. Additional experiment details

## A.1. Details on prompt engineering

Accordindg to the prompt engineering techniques described in (Schulhoff et al., 2024), we design our prompts in the following ways. The first prompt engineering techinique we adopted three prompting techniques: (1) role-based prompting, (2) self-consistency prompting, and (3) contrastive prompting.

**Role-based prompting**. For the role-based prompting, our prompt is ""You are a medical assistant trained to assess Alzheimer's Disease (AD) risk using both structured patient data and natural language transcripts of picture description tasks. Given a patient's demographic information, two NLP-derived categories (lexical diversity and filler word usage), and their transcript, decide whether they are likely to have Alzheimer's Disease (label as 1) or not (label as 0). Your prediction should rely primarily on semantic coherence, richness of description, sentence structure, and topic maintenance in the transcripts, supported by the NLP-derived categories.Focus especially on: (1) Semantic richness: Are objects and actions well described? (2) Syntactic fluency: Are sentences grammatically structured? (3) Discourse organization: Is the description cohesive, progressing logically? (4) Error patterns: Do you see word-finding difficulty, repetition, or vagueness? Interpretation of NLP-derived categories: (1) Lexical Diversity Category (Type-Token Ratio): High lexical diversity: Typical of cognitively healthy individuals. Moderate lexical diversity: Mildly reduced; possibly normal aging. Low-moderate lexical diversity: Suggestive of mild cognitive impairment. Low lexical diversity: Often indicative of Alzheimer's or dementia. (2) Filler Word Usage Category: Normal: Typical cognitive function. Mildly elevated: Possible mild cognitive changes. Elevated: Indicative of cognitive impairment or Alzheimer's. (3) Patient Information: Sex: Sex. Race: Race. Education (years): Education. Lexical Diversity Category: ttr category. Filler Word Usage Category: filler ratio category. Speech excerpt: Transcript Return your answer as 0 (unlikely AD) or 1 (likely AD). No explanation is needed."

**Self-consistency prompting** For the self-consistency prompting, our prompt is: "You are a medical assistant trained to assess Alzheimer's Disease (AD) risk using structured patient data and natural language transcripts of picture description tasks. Given the patient information and transcript below, evaluate Alzheimer's Disease likelihood through three separate analyses: Analysis 1 - Semantic and Syntactic Quality: - Semantic richness: Are objects and actions clearly and accurately described? - Syntactic fluency: Are sentences grammatically structured? Provide an initial assessment: 0 (unlikely AD) or 1 (likely AD). Analysis 2 - Discourse and Error Patterns: - Discourse organization: Does the description logically progress and remain coherent? - Error patterns: Are there noticeable word-finding difficulties, repetition, or vagueness? Provide an initial assessment: 0 (unlikely AD) or 1 (likely AD). Analysis 3 - NLP-derived Categories: Lexical Diversity Category (Type-Token Ratio): - High lexical diversity: Typical cognitive health. - Moderate lexical diversity: Possibly normal aging. - Low-moderate lexical diversity: Suggestive of mild impairment. - Low lexical diversity: Indicative of Alzheimer's/dementia. Filler Word Usage Category: - Normal: Typical cognitive function. - Mildly elevated: Possible mild cognitive changes. - Elevated: Indicative of cognitive impairment/Alzheimer's. Provide an initial assessment: 0 (unlikely AD) or 1 (likely AD). Patient Information: - Sex: Sex - Race: Race - Education (years): Education - Lexical Diversity Category: ttr category - Filler Word Usage Category: filler ratio category - Speech excerpt: Transcript. Based on your three analyses above, provide your final prediction by taking a majority vote: 0 (unlikely AD) or 1 (likely AD). Final Answer:"

**Contrastive prompting**. For the contrasive prompting technique: "You are a medical assistant trained to assess Alzheimer's Disease (AD) risk using structured patient data and natural language transcripts of picture description tasks. Consider the following two example patient excerpts carefully and note their linguistic differences:

Example 1 (Likely AD): - Low semantic richness (limited object/action description). - Poor syntactic fluency (frequent grammatical mistakes). - Poor discourse organization (disjointed, incoherent description). - Frequent error patterns (word-finding difficulty, repetition). - Lexical Diversity Category: Low - Filler Word Usage Category: Elevated

Example 2 (Unlikely AD): - High semantic richness (detailed object/action description). - Good syntactic fluency (grammatically correct sentences). - Strong discourse organization (logical, coherent description). - Few or no error patterns. - Lexical Diversity Category: High - Filler Word Usage Category: Normal

Now, using these contrasts as reference, assess the following patient's transcript:

Patient Information: - Sex: Sex - Race: Race - Education (years): Education - Lexical Diversity Category: ttr category - Filler Word Usage Category: filler ratio category - Speech excerpt: Transcript

Classify this patient as 0 (unlikely AD) or 1 (likely AD) based on similarity to Example 1 or Example 2. Provide your answer without additional explanation.

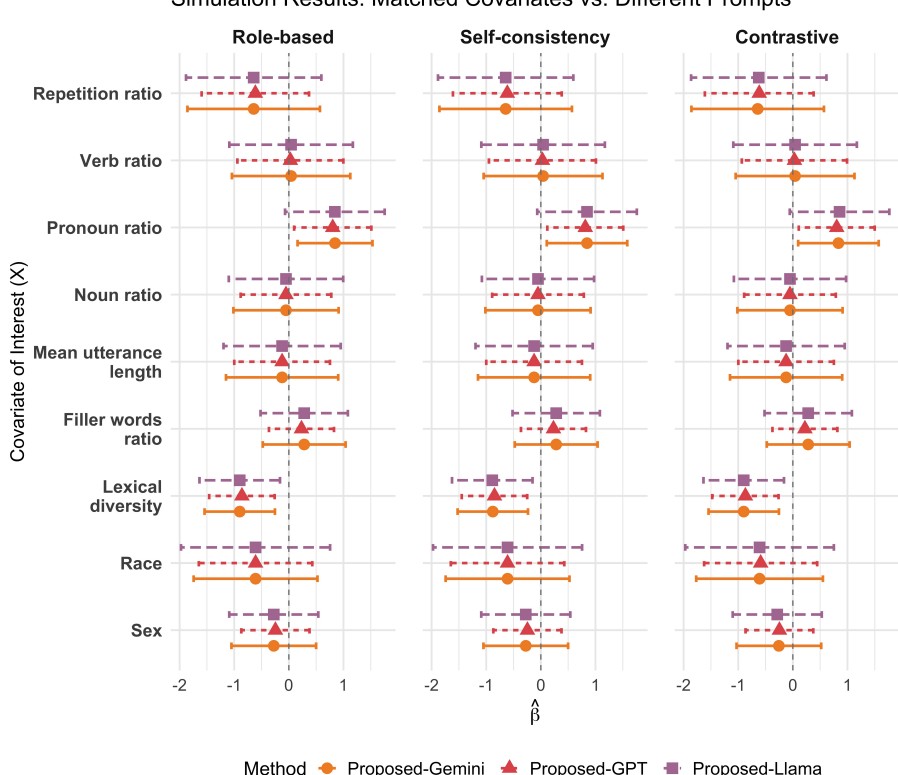

*Figure 3.* Comparison of simulation performance under different prompts

Final Answer:"

### A.2. Details on YouTube data extraction pipeline and structured linguistic feature extraction

**Brief YouTube data extraction pipeline**. Our approach queries Youtube videos involved the combination across the two categories of keywords: primary keywords: "Alzheimer's disease patient," "Dementia patient", and secondary context-specific keywords: "MMSE," "Cognitive test," "Interview," "Caregiver," "Family interaction," "Therapy sessions," "Doctor". For each keyword combination, we retrieve up to 70 videos that have no licensing restrictions and the best available video quality in .mp4 format. Comments associated with each video are separately collected and stored in individual text file format. Furthermore, metadata—including titles, descriptions, viewer engagement metrics, licensing types, and video tags—are compiled into structured tabular format. This targeted keyword methodology ensures a comprehensive coverage of relevant AD-related content across diverse interaction contexts. To extract unstructured, each queried YouTube video was analyzed using Gemini models to systematically extract detailed qualitative data.

We adopt a structured prompt specifically designed to comprehensively capture critical qualitative dimensions of dementia patient interactions. The structured prompt for YouTube data extraction includes the following components: (1) participant overview, enumerating the total number of unique individuals visible or audible, along with their roles (e.g., patient, caregiver, interviewer); (2) patient vocal presence; (3) interaction patterns among participants (e.g., caregiver-patient, interviewer-patient); (4) evidence of cognitive impairment, including explicit quotes describing impairment; (5) the patient's emotional state (e.g., calm, anxious, frustrated); (6) documentation of any unusual behaviors (e.g., aggression, wandering, confusion); (7) the patient's ability to perform daily activities; (8) caregiver roles, specifying assistance provided; (9) details of patient interactions with medical professionals; (10) presence and nature of language difficulties (e.g., slurred speech, word-finding issues); (11) observations on the use of body language; (12) speech characteristics, noting pauses or hesitations; (13) a precise dialogue summary, clearly transcribed according to speaker roles; (14) patient voice-transcript pairs marked with accurate timestamps for each continuous patient utterance; and (15) identification of recurring comment themes from viewer interactions. This structured and detailed prompt enabled consistent, accurate extraction of rich qualitative and

behavioral insights, significantly enhancing the dataset's readiness for subsequent advanced multimodal analyses.

**Structured linguistic feature extraction**. To extract structured linguistic features, we employ natural language processing (NLP) techniques to derive quantitative linguistic features from each transcript. We analyze each transcript and extract a range of NLP features, including total word count, number of unique words, type-token ratio (vocabulary richness), mean utterance length, and ratios of specific parts of speech (nouns, pronouns, verbs). Furthermore, we quantify filler word usage and identify the most frequent words within transcripts. The NLP feature extraction was applied consistently across all collected transcripts, resulting in structured datasets enriched with linguistic metrics. These NLP-derived features were integrated into the metadata, further enhancing the dataset's analytical potential and supporting advanced quantitative analyses in combination with qualitative insights.

### A.3. Additional experiment results

In this section, we provide additional simulation results.

**Comparison with PPI-based methods.** As shown in Table 2, PPI+LLM improves efficiency over standard PPI, while LASS achieves the largest gain. The column "Adaptability to Unstructured Data," indicating which methods can directly handle unstructured features.

| Covariate | Method | | | |
|---|---|---|---|---|
| | **Labeled data only** | **LASS** | **PPI** | **PPI+LLM** |
| Lexical diversity | 0.74 | 0.52 | 0.62 | 0.58 |
| Race | 0.71 | 0.58 | 0.59 | 0.60 |
| Sex | 0.80 | 0.65 | 0.69 | 0.67 |
| Pronoun ratio | 0.76 | 0.56 | 0.63 | 0.60 |
| Filler words ratio | 0.60 | 0.43 | 0.44 | 0.43 |
| Verb ratio | 0.56 | 0.38 | 0.41 | 0.37 |
| Adaptability to unstructured data | $\times$ | $\checkmark$ | $\times$ | $\checkmark$ |

*Table 2.* Comparison of PPI-based methods across covariates

**Effect of different prompting techniques.** To evaluate the performance of our methods under different prompting techniques, we compare three prompt engineering methods to generate the predicted outcomes using LLMs: (1) role-based prompting, which instructs the LLM by explicitly defining its role or persona to guide its reasoning and responses, (2) self-consistency prompting, which generates multiple responses from a single prompt and aggregates them to enhance prediction robustness, and (3) contrastive prompting, which simultaneously queries both positive and negative prompts and compares responses to refine prediction accuracy. We first demonstrate the effect of different prompting techniques on the experimental results in Figure 3. Furthermore, we provide experimental results under matched covariates for $n + N = 600$, where $n/N = 0.17$ in Figure 4. Figure 3 suggests that our proposed methods are not sensitive to the choice of prompting techniques. The results in Figure 4 show the simulation results of our proposed method under a smaller sample size. Our proposed method shows higher estimation efficiency compared with the other benchmark methods, which aligns with our simulation results in the main manuscript.

**Effect of ill-designed prompts.** To assess the impact of ill-designed prompts, we report additional results in Table 3 for two prompt specifications: (i) an unfixed prompt and (ii) a prompt originally designed for a different task. For the latter, we leverage the additional IMDb case study included in our paper to construct a mismatched prompt that is not tailored to the linguistic and demographic prediction task considered here. The results show that, under both ill-designed prompts, the efficiency gains of LASS are substantially weakened and become much closer to the labeled-data-only baseline. This finding further supports the methodological intuition underlying LASS: when prompts are poorly specified or mismatched to the target task, the resulting pseudo-outcomes become less informative, and the corresponding inferential advantage diminishes.

**Robustness to LLM hallucination.** A variety of other strategies have been explored to detect or mitigate hallucinations in LLMs: retrieval-augmented generation (Feldman et al., 2023; Zhang et al., 2024; Peng et al., 2023; Varshney et al., 2023), custom token sampling procedures (Chuang et al., 2023; Liu et al., 2024; Huang et al., 2025) as well as learning to extract or

| Covariate | Method | | | |
|---|---|---|---|---|
| | Labeled data only | LASS (unfixed) | LASS (prompt for different task) | LASS |
| Lexical diversity | 0.74 | 0.72 | 0.72 | 0.52 |
| Race | 0.71 | 0.70 | 0.69 | 0.58 |
| Sex | 0.80 | 0.78 | 0.79 | 0.65 |
| Pronoun ratio | 0.76 | 0.75 | 0.75 | 0.60 |
| Filler words ratio | 0.60 | 0.58 | 0.56 | 0.43 |
| Verb ratio | 0.56 | 0.55 | 0.55 | 0.38 |

*Table 3.* Summary of standard deviations across six linguistic and demographic variables under ill-designed prompts.

steer truthfulness from hidden states (Burns et al., 2022).

To minimize the effect of hallucination, we use structured prompts that explicitly constrain the LLM output format. To further validate this design choice, we compare our proposed method under the structured prompt with an alternative variant using unstructured prompts without hallucination control. As summarized in the following table, the version without structured prompts "LASS (hallucination)" yields standard deviations comparable to using labeled data only, meaning the LLM predictions do not assist with efficiency improvement. The results suggest that our hallucination control is necessary in ensuring the quality of LLM predictions.

| Covariate | Method | | | |
|---|---|---|---|---|
| | Labeled data only | LASS (unfixed) | LASS ($T$=0.5) | LASS |
| Lexical diversity | 0.74 | 0.72 | 0.67 | 0.52 |
| Race | 0.71 | 0.70 | 0.68 | 0.58 |
| Sex | 0.80 | 0.78 | 0.75 | 0.65 |
| Pronoun ratio | 0.76 | 0.75 | 0.71 | 0.60 |
| Filler words ratio | 0.60 | 0.58 | 0.52 | 0.43 |
| Verb ratio | 0.56 | 0.55 | 0.47 | 0.38 |

*Table 4.* Summary of standard deviations across six linguistic and demographic variables.

Furthermore, we provide additional simulation results in Table 5. We have now extended the temperature analysis to $T \in \{0, 0.1, 0.2, 0.3, 0.5, 0.7, 0.9\}$. The additional results for $T < 0.5$ show that, as $T$ decreases from 0.5 to 0, the standard deviation decreases across all six covariates, suggesting improved estimation efficiency at lower temperatures. This pattern indicates that temperature acts as a noise parameter in the LLM-generated pseudo-outcome, which is consistent with our theoretical intuition. When the temperature is very small, such as $T = 0.1$, the highest-probability label is still selected most of the time; consequently, the results for $T = 0.1$ are already close to those for $T = 0$. In contrast, larger temperatures introduce more sampling randomness and change the predicted label more frequently, weakening the informativeness of the pseudo-outcomes and leading to the larger standard deviations observed for $T > 0.5$.

| Covariate | Labeled data only | LASS ($T$=0.9) | LASS ($T$=0.7) | LASS ($T$=0.5) | LASS ($T$=0.3) | LASS ($T$=0.2) | LASS ($T$=0.1) | LASS ($T$=0) |
|---|---|---|---|---|---|---|---|---|
| Lexical diversity | 0.74 | 0.72 | 0.70 | 0.67 | 0.64 | 0.58 | 0.52 | 0.52 |
| Race | 0.71 | 0.69 | 0.69 | 0.68 | 0.66 | 0.65 | 0.60 | 0.58 |
| Sex | 0.80 | 0.78 | 0.77 | 0.75 | 0.71 | 0.70 | 0.69 | 0.65 |
| Pronoun ratio | 0.76 | 0.74 | 0.74 | 0.73 | 0.67 | 0.65 | 0.61 | 0.60 |
| Filler words ratio | 0.60 | 0.59 | 0.57 | 0.52 | 0.49 | 0.48 | 0.46 | 0.43 |
| Verb ratio | 0.56 | 0.51 | 0.48 | 0.47 | 0.43 | 0.42 | 0.40 | 0.38 |

*Table 5.* Summary of standard deviations across six linguistic and demographic variables under different temperatures. Smaller values indicate greater efficiency.

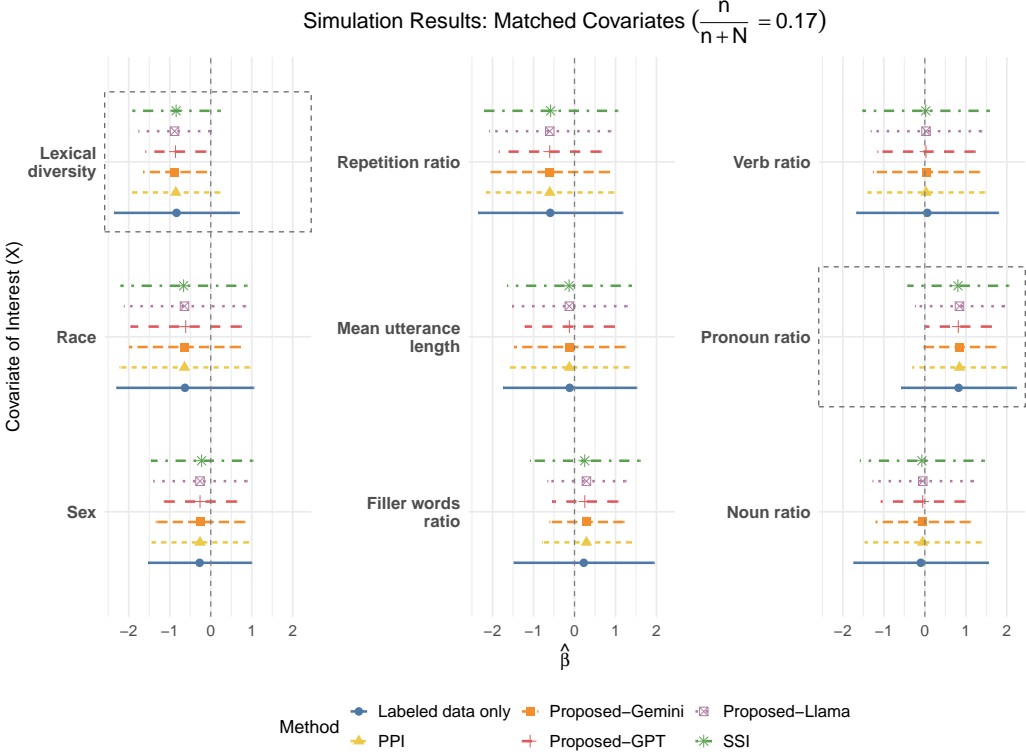

*Figure 4.* Comparison of our proposed method and the benchmark methods for various covariates.

**Our proposed method using CART versus random forest.** Practically, our framework can adopt more stable segmentation strategies—such as growing a full tree followed by a PIR-specific trimming technique, optimal trees, or random forests—without compromising validity. An additional comparison of LASS using CART versus random forests is presented in Table 6, which shows that the random-forest variant consistently yields smaller standard deviations, indicating improved stability from ensembling:

| Covariate | Proposed+CART | Proposed+RF |
|---|---|---|
| Lexical diversity | 0.35 | 0.32 |
| Race | 0.56 | 0.55 |
| Sex | 0.61 | 0.59 |
| Pronoun ratio | 0.48 | 0.48 |
| Filler words ratio | 0.36 | 0.35 |
| Verb ratio | 0.30 | 0.28 |

*Table 6.* Comparison of standard deviations across six linguistic and demographic covariates for two variants (CART, random forest) of the prediction invariant region selection.

## B. Additional real world case study results

In this section, we present additional real world case study results. For the first case study using the real DementiaBank dataset and the self-extracted YouTube video data, we apply our proposed method to unmatched covariates to uncover scientific insights and present our case study results in Figure 5. We present the prediction-invariant region in Figure 6. Figure 5 suggests that our proposed method identifies lexical diversity and pronoun ratio as statistically significant covariates for predicting AD status.

For the second case study using IMDb data, we summarize our results in Table 7. The IMDb dataset contains 25,000 labeled

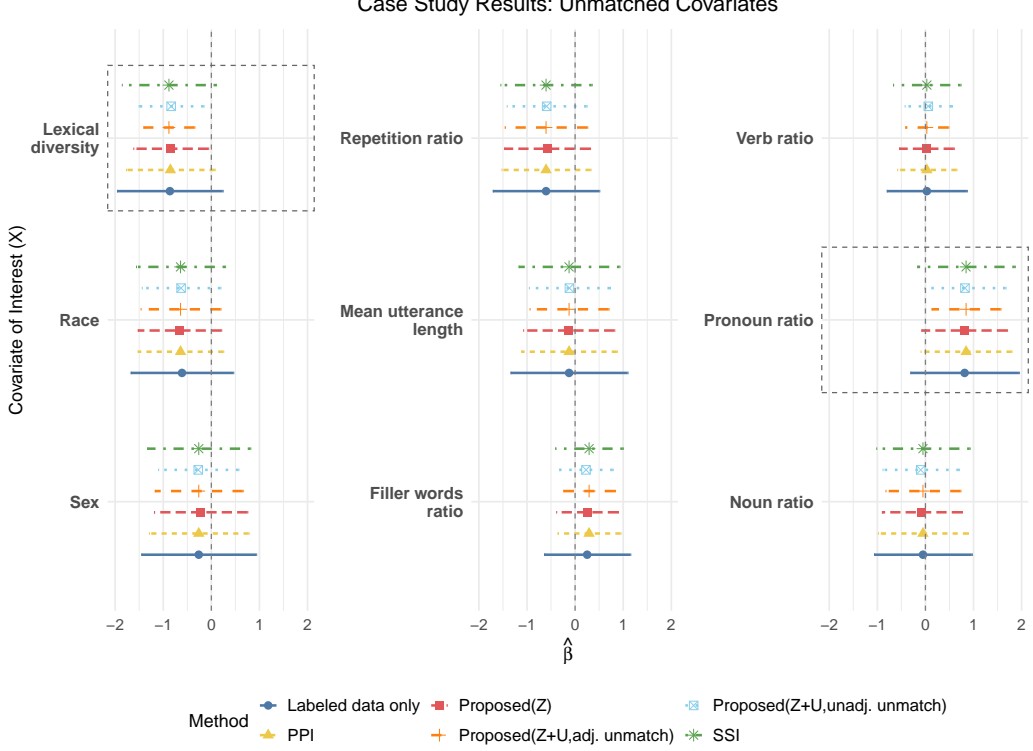

*Figure 5.* Case study results (unmatched covariates).

training reviews and 25,000 labeled test reviews, along with an additional 3.4 million unlabeled IMDb reviews, making it particularly suitable for semi-supervised and LLM-augmented inference tasks. The outcome of interest is the binary sentiment label. The structured covariates include features of the reviews, such as length, type-token ratio, and negation counts. The unstructured covariates include the raw review texts. The results suggest that our proposed method yields narrower confidence intervals than the benchmark methods, indicating improved estimation efficiency and confirming that the same performance pattern extends to this IMDb dataset.

| Method | Length | Token ratio | Negation counts |
|---|---|---|---|
| Labeled data only | $[-0.15,\ 0.17]$ | $[-0.16,\ 0.22]$ | $[-0.41,\ 0.03]$ |
| SSI | $[-0.13,\ 0.15]$ | $[-0.14,\ 0.19]$ | $[-0.38,\ 0.00]$ |
| PPI | $[-0.13,\ 0.15]$ | $[-0.13,\ 0.18]$ | $[-0.36,\ -0.02]$ |
| LASS | $[-0.11,\ 0.13]$ | $[-0.13,\ 0.17]$ | $[-0.34,\ -0.03]$ |

*Table 7.* Comparison of confidence intervals with LASS and the benchmark methods applied on the IMDb dataset.

## C. Proof of Theorem 1

### C.1. Consistency

Besides Assumption 4.1, we impose standard regularity conditions for GLMs to ensure the validity of asymptotic arguments:

**Assumption C.1** (Parameter space and smoothness). The true parameter $\beta^*$ lies in a compact set $\mathcal{B} \subset \mathbb{R}^p$, and the link function $g(x^\top \beta)$ is continuously differentiable in $\beta$ for all covariates $x \in \mathcal{X}$.

**Assumption C.2** (Identifiability). The true parameter $\beta^* \in \mathcal{B}$ is the unique solution to the population moment equation:
$$\Psi(\beta) = \mathbb{E}\left[ X\left( Y - g\left( X^\top \beta \right) \right) \right] = 0.$$

To simplify the exposition, we introduce an indicator variable $S \in \{0, 1\}$ denoting the data source: $S_i = 1$ if subject $i$

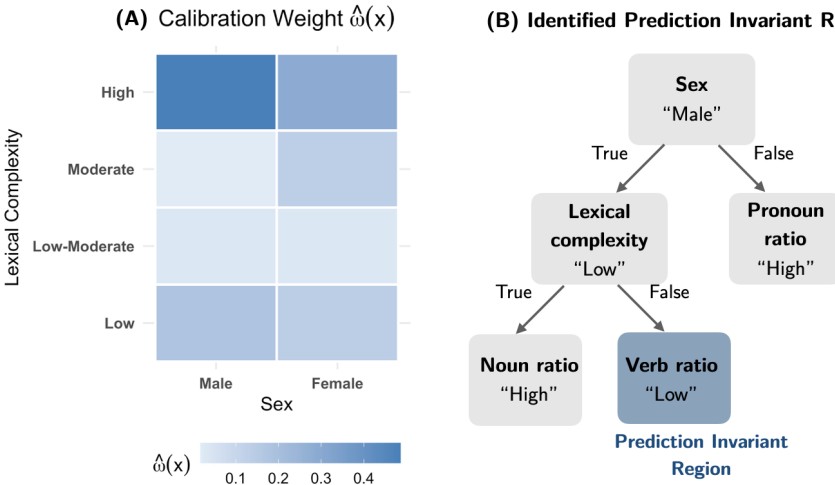

*Figure 6.* Prediction-invariant region and calibration weights identified in real world data.

is labeled (i.e., $Y_i$ is observed), and $S_i = 0$ otherwise. The observed structured data with LLM prediction is denoted by $O = (X, Y, Y^*, S)$. The influence function is defined as:

$$\psi(O; \beta, \eta, \pi) = X\{[m(X) + \omega(X)(Y^* - m^*(X))](1 - S/\pi)$$
$$+ SY/\pi - g(X^\top\beta)\},$$

where $\eta = (m, m^*, \omega)$ denotes the collection of nuisance functions and $\pi = \mathbb{P}(S = 1)$ is the labeling probability. The estimator $\hat{\beta}_{\text{LASS}, \text{xz}}$ solves the solution to the empirical estimating equation:

$$\hat{\Psi}_{N+n}(\beta) = \frac{1}{N+n}\sum_{i=1}^{N+n}\psi(O_i; \beta, \hat{\eta}, \hat{\pi}) = 0,$$

where $\hat{\pi} = \frac{n}{N+n} = \pi + O_p\left(\frac{1}{\sqrt{N+n}}\right)$. Further define the empirical estimation equation with true nuisance parameters as:

$$\Psi_{N+n}(\beta) = \frac{1}{N+n}\sum_{i=1}^{N+n}\psi(O_i; \beta, \eta, \pi).$$

It follows from Assumptions 4.1.2 and 4.1.4 that

$$\sup_{\beta \in B}\left\|\hat{\Psi}_{N+n}(\beta) - \Psi_{N+n}(\beta)\right\| = o_p(1).$$

Moreover, since $\mathbb{E}[\psi(O; \beta, \eta, \pi)] = \Psi(\beta)$ and the class of functions $\{\psi(O; \beta, \eta, \pi) : \beta \in \mathcal{B}\}$ admits an integrable envelope, the class is Glivenko-Cantelli by Assumptions 4.1.2 and C.1. Hence,

$$\sup_{\beta \in \mathcal{B}}\|\Psi_{N+n}(\beta) - \Psi(\beta)\| = o_p(1).$$

Combining the above results yields

$$\sup_{\beta \in \mathcal{B}}\left\|\hat{\Psi}_{N+n}(\beta) - \Psi(\beta)\right\| \leqslant \sup_{\beta \in \mathcal{B}}\|\Psi_{N+n}(\beta) - \Psi(\beta)\|$$
$$+ \sup_{\beta \in \mathcal{B}}\left\|\hat{\Psi}_{N+n}(\beta) - \Psi_{N+n}(\beta)\right\| = o_p(1).$$

Therefore, under Assumption 4.1, C.1, and C.2, we apply Theorem 5.9 of (Van der Vaart, 2000) to conclude that $\hat{\beta}_{\text{LASS}, \text{xz}} \xrightarrow{P} \beta^*$.

## C.2. Asymptotic normality

We follow the standard derivation strategy outlined in Chapter 5 of (Van der Vaart, 2000). To establish asymptotic normality, we expand the estimating equation $\hat{\Psi}_{N+n}(\hat{\beta}_{\mathtt{LASS,xz}})$ around the true parameter value $\beta^*$ using a Taylor expansion. For simplicity, we temporarily suppress the subscript $\mathtt{LASS,xz}$ and write:

$$0 = \hat{\Psi}_{N+n}(\hat{\beta}) = \hat{\Psi}_{N+n}(\beta^*) + \nabla_\beta \hat{\Psi}_{N+n}(\beta^*)(\hat{\beta} - \beta^*)$$
$$+ \frac{1}{2}(\hat{\beta} - \beta^*)^\top \nabla_\beta^2 \hat{\Psi}_{N+n}(\beta^*)(\hat{\beta} - \beta^*),$$

where the Jacobian $\nabla_\beta \hat{\Psi}_{N+n}(\beta^*) = \left[\partial \hat{\Psi}_{N+n,r}/\partial \theta_s\right]_{r,s=1}^p \in \mathbb{R}^{p \times p}$, the Hessian $\nabla_\beta^2 \hat{\Psi}_{N+n}(\beta^*) = \left[\partial^2 \hat{\Psi}_{N+n,r}/\partial \theta_s \partial \theta_t\right]_{r,s,t=1}^p$ is a third-order tensor. Rearranging terms, we obtain:

$$\sqrt{N+n}(\hat{\beta} - \beta^*) = - \underbrace{\left(\nabla_\beta \hat{\Psi}_{N+n}(\beta^*) + \frac{1}{2}(\hat{\beta} - \beta^*)^\top \nabla_\beta^2 \hat{\Psi}_{N+n}(\beta^*)\right)^{-1}}_{\text{(I)}} \tag{6}$$
$$\cdot \underbrace{\sqrt{N+n}\,\hat{\Psi}_{N+n}(\beta^*)}_{\text{(II)}}$$

Since $\hat{\beta} \xrightarrow{p} \beta^*$ by consistency, we can expand (I):

$$\text{(I)} = \nabla_\beta \hat{\Psi}_{N+n}(\beta^*) + o_p(1) = -\frac{1}{N+n}\sum_{i=1}^{N+n} g'(X_i^\top \beta^*) X_i X_i^\top + o_p(1)$$
$$= -\mathbb{E}\left[g'\left(X^\top \beta^*\right) X X^\top\right] + o_p(1).$$

To analyze the asymptotic behavior of term (II), we require additional conditions on the estimated nuisance functions $\hat{\eta} = (\hat{m}, \hat{m}^*, \hat{\omega})$.

**Assumption C.3** (Nuisance functions). At least one of the following conditions holds:

1. **Donsker Condition.** The class of functions $\{\psi(O; \beta, \eta, \pi) : \eta \in \mathcal{M}_{N+n}\}$ is Donsker, where $\mathcal{M}_{N+n}$ is a random neighborhood of the true nuisance parameter $\eta$ that contains the estimated function $\hat{\eta}$ with high probability.

2. **Cross-Fitting.** The nuisance function estimator $\hat{\eta}(X_i)$ for subject $i$ is obtained using sample folds that exclude subject $i$, i.e., $\hat{\eta}(X_i)$ is learned on data independent of $O_i$.

The Donsker condition, which ensures asymptotic equicontinuity and uniform convergence, is discussed in detail in Chapters 5.4 and 19 of (Van der Vaart, 2000). However, this condition can be restrictive in high-dimensional or complex settings common in modern applications. To address this limitation, one can adopt cross-fitting strategies, as proposed in (Chernozhukov et al., 2018), which allow for flexible machine learning-based nuisance estimation while retaining valid asymptotic properties.

**Lemma C.4.** *Under Assumption 4.1 and C.1-C.3, we have:*

$$\sqrt{N+n}\left(\hat{\Psi}_{N+n}(\beta^*) - \Psi_{N+n}(\beta^*)\right) = o_p(1).$$

This result implies that the stochastic fluctuation of the estimator is primarily driven by $\Psi_{N+n}(\beta^*)$, the population-level estimating equation evaluated at the true parameter. To characterize its asymptotic behavior and illustrate the role of the optimal calibration weight $\omega^*(X)$, we begin by analyzing the general case with arbitrary weighting function $\omega(X)$. The asymptotic variance of $\sqrt{N+n}\,\Psi_{N+n}(\beta^*)$ can be decomposed into three components:

$$B = \text{Asymp.}\,\text{Var}\left(\sqrt{N+n}\,\Psi_{N+n}(\beta^*)\right) = V_1 + V_2 + 2V_{12}, \tag{7}$$

where,

$$V_1 = \mathbb{E}\left\{ \left[ \omega(X)Y^* + \frac{S}{\pi}\left(Y - \omega(X)Y^*\right) - g(X^\top \beta^*) \right]^2 XX^\top \right\}$$

$$= \mathbb{E}\left\{ \left[ \left(Y - g(X^\top \beta^*)\right)^2 + (1/\pi - 1)\left(Y - \omega(X)Y^*\right)^2 \right] XX^\top \right\},$$

where the second equality uses the moment condition $\mathbb{E}[YX] = \mathbb{E}[g(X^\top \beta^*)X]$ implied by the estimating equation. Similarly, the second variance component is

$$V_2 = \mathbb{E}\left\{ \left[ (m(X) - \omega(X)m^*(X))\left(\frac{S}{\pi} - 1\right) \right]^2 XX^\top \right\}$$

$$= \mathbb{E}\left\{ (m(X) - \omega(X)m^*(X))^2 \left(\frac{1}{\pi} - 1\right) XX^\top \right\},$$

and the covariance term is given by

$$V_{12} = -\mathbb{E}\left\{ \left[ \omega(X)Y^* + \frac{S}{\pi}\left(Y - \omega(X)Y^*\right) - g(X^\top \beta^*) \right] \right.$$

$$\left. \cdot (m(X) - \omega(X)m^*(X))\left(\frac{S}{\pi} - 1\right) XX^\top \right\}$$

$$= -\mathbb{E}\left\{ (m(X) - \omega(X)m^*(X))^2 \left(\frac{1}{\pi} - 1\right) XX^\top \right\},$$

where we use conditional independence and mean-zero cross terms. Substituting $V_1$, $V_2$, and $V_{12}$ into Equation (7), we obtain the final expression for the asymptotic variance:

$$B = V_1 + V_2 + 2V_{12}$$

$$= \mathbb{E}\left[ \left(Y - g(X^\top \beta^*)\right)^2 XX^\top \right]$$

$$+ \mathbb{E}\left\{ \left(\frac{1}{\pi} - 1\right) \mathrm{Var}\left(Y - \omega(X)Y^* \mid X\right) XX^\top \right\}.$$

If we choose the weighting function to be:

$$\omega(X) = \omega^*(X) = \frac{\mathbb{C}\mathrm{ov}(Y, Y^* \mid X = x)}{\mathbb{V}(Y^* \mid X = x)},$$

then the asymptotic variance $\mathbb{V}\left(\sqrt{N+n}\,\hat{\Psi}_{N+n}(\beta^*)\right)$ is minimized over all choices of $\omega(\cdot)$. Under this optimal choice, the asymptotic variance simplifies to:

$$B = \mathbb{E}\left[ \left(Y - g\left(X^\top \beta\right)\right)^2 XX^\top \right]$$

$$+ \mathbb{E}\left[ XX^\top \left(\frac{1}{\pi} - 1\right)\left(\mathrm{Var}(Y \mid X) - \frac{\mathrm{Cov}^2\left(Y^*, Y \mid X\right)}{\mathrm{Var}\left(Y^* \mid X\right)}\right) \right].$$

To further interpret the results, note that:

$$\mathbb{E}\left[ \left(Y - g(X^\top \beta)\right)^2 XX^\top \right] = \mathbb{E}\left[ XX^\top \,\mathrm{Var}(Y \mid X) \right]$$

$$+ \mathbb{E}\left[ XX^\top \left(m(X) - g(X^\top \beta)\right)^2 \right].$$

Substituting this decomposition into the expression for $B$, we obtain:

$$B = \mathbb{E}\left[ \frac{1}{\pi}\left(Y - g(X^\top \beta)\right)^2 XX^\top \right]$$

$$- \mathbb{E}\left[ \left(\frac{1}{\pi} - 1\right)\left(\left(m(X) - g(X^\top \beta)\right)^2 + \frac{\mathrm{Cov}^2(Y^*, Y \mid X)}{\mathrm{Var}(Y^* \mid X)}\right) XX^\top \right].$$

Letting $A := \mathbb{E}\left[g'\left(X^\top \beta^*\right) XX^\top\right]$, and applying the multivariate CLT along with Slutsky's theorem to the expansion in Equation (6), we conclude:

$$\sqrt{N+n}\left(\hat{\beta}_{\texttt{LASS,xz}} - \beta^*\right) \xrightarrow{d} \mathcal{N}(0, A^{-1}BA^{-1}).$$

In practice, the optimal weighting function $\omega^*(X)$ may be difficult to estimate precisely. A practical solution is to approximate $\omega(X)$ using a coarsened version of the covariate $X$, such as by discretizing $X$ into bins. While this introduces a slight loss in statistical efficiency, the resulting estimator remains asymptotically normal.

*Proof of Lemma C.4.* We prove the result under the cross-fitting scheme using $K$ folds $\{\mathcal{D}_k\}_{k=1}^K$. Let the cross-fitted nuisance estimator for subject $i \in \mathcal{D}_k$ be denoted by $\hat{\eta}^{-k}(X_i)$, which is trained on data excluding fold $k$. We begin by decomposing the difference between the empirical and pseudo-population estimating functions:

$$\hat{\Psi}_{N+n}(\beta^*) - \Psi_{N+n}(\beta^*) = \frac{1}{N+n}\sum_{i=1}^{N+n}\left[\psi(O_i; \beta^*, \hat{\eta}, \hat{\pi}) - \psi(O_i; \beta^*, \hat{\eta}, \pi)\right]$$

$$+ \frac{1}{N+n}\sum_{i=1}^{N+n}\left[\psi(O_i; \beta^*, \hat{\eta}, \pi) - \psi(O_i; \beta^*, \eta, \pi)\right]. \tag{8}$$

We first consider the second term. Using the cross-fitting structure, we write:

$$\frac{1}{N+n}\sum_{i=1}^{N+n}\left[\psi(O_i; \beta^*, \hat{\eta}, \pi) - \psi(O_i; \beta^*, \eta, \pi)\right] =$$

$$\sum_{k=1}^K \frac{|\mathcal{D}_k|}{N+n} \cdot \frac{1}{|\mathcal{D}_k|}\sum_{i \in \mathcal{D}_k}\left[\psi(O_i; \beta^*, \hat{\eta}^{-k}, \pi) - \psi(O_i; \beta^*, \eta, \pi)\right].$$

Hence, it suffices to control the convergence rate within each fold $k$. Consider the decomposition:

$$\frac{1}{|\mathcal{D}_k|}\sum_{i \in \mathcal{D}_k}\left[\psi(O_i; \beta^*, \hat{\eta}^{-k}, \pi) - \psi(O_i; \beta^*, \eta, \pi)\right]$$

$$= \frac{1}{|\mathcal{D}_k|}\sum_{i \in \mathcal{D}_k}\left(1 - \frac{S_i}{\pi}\right)X_i \cdot \left[\hat{m}^{-k}(X_i) - m(X_i)\right.$$

$$+ \left(\hat{\omega}^{-k}(X_i) - \omega^*(X_i)\right)(Y_i^* - m^*(X_i))$$

$$+ \left(\hat{m}^{*-k}(X_i) - m^*(X_i)\right)\omega^*(X_i)$$

$$+ \left.\left(\hat{\omega}^{-k}(X_i) - \omega^*(X_i)\right)\left(\hat{m}^{*-k}(X_i) - m^*(X_i)\right)\right].$$

We now analyze the first term. By conditioning on $\mathcal{D}_{-k}$ and $\{X_i\}_{i \in \mathcal{D}_k}$, and treating $\hat{\eta}^{-k}$ as fixed, we have:

$$\text{Var}\left(\frac{1}{|\mathcal{D}_k|}\sum_{i \in \mathcal{D}_k}\left(1 - \frac{S_i}{\pi}\right)\left(\hat{m}^{-k}(X_i) - m(X_i)\right)X_i \, \middle| \, \mathcal{D}_{-k}, \{X_i\}_{i \in \mathcal{D}_k}\right)$$

$$= \frac{1}{|\mathcal{D}_k|^2}\sum_{i \in \mathcal{D}_k}\mathbb{E}\left[\left(\hat{m}^{-k}(X_i) - m(X_i)\right)^2\left(1 - \frac{S_i}{\pi}\right)^2 X_i X_i^\top\right]$$

$$= \frac{1-\pi}{\pi} \cdot \frac{1}{|\mathcal{D}_k|^2}\sum_{i \in \mathcal{D}_k}\left(\hat{m}^{-k}(X_i) - m(X_i)\right)^2 X_i X_i^\top$$

$$= o_p\left(\frac{1}{N+n}\right).$$

Then by Chebyshev's inequality,

$$\frac{1}{|\mathcal{D}_k|}\sum_{i \in \mathcal{D}_k}\left(1 - \frac{S_i}{\pi}\right)X_i\left(\hat{m}^{-k}(X_i) - m(X_i)\right) = o_p\left(\frac{1}{\sqrt{N+n}}\right).$$

Similar arguments apply to the remaining three terms. Thus, the overall contribution from this difference is:

$$\frac{1}{N+n} \sum_{i=1}^{N+n} [\psi(O_i; \beta^*, \hat{\eta}, \pi) - \psi(O_i; \beta^*, \eta, \pi)] = o_p \left( \frac{1}{\sqrt{N+n}} \right).$$

Next, we handle the first term in the decomposition (8), involving the difference in estimated and true labeling probabilities:

$$\frac{1}{N+n} \sum_{i=1}^{N+n} [\psi(O_i; \beta^*, \hat{\eta}, \hat{\pi}) - \psi(O_i; \beta^*, \hat{\eta}, \pi)]$$

$$= \frac{1}{n} (\hat{\pi} - \pi) \sum_{i=N+1}^{N+n} \frac{1}{\pi} [Y_i - \hat{m}(X_i) - \hat{\omega}(X_i)(Y_i^* - \hat{m}^*(X_i))]$$

$$= \frac{1}{n} (\hat{\pi} - \pi) \sum_{i=N+1}^{N+n} \frac{1}{\pi} [Y_i - m(X_i) - \omega^*(X_i)(Y_i^* - m^*(X_i))]$$

$$+ o_p \left( \frac{1}{\sqrt{N+n}} \right)$$

$$= o_p \left( \frac{1}{\sqrt{N+n}} \right),$$

where the second equality uses the consistency of $\hat{\eta}$ and the expansion $\hat{\pi} = \pi + O_p \left( \frac{1}{\sqrt{N+n}} \right)$. This step does not require cross-fitting and follows directly from convergence of $\hat{\pi}$ and the stability of the estimating function. Combining both terms, we conclude from (8) that $\hat{\Psi}_{N+n}(\beta^*) - \Psi_{N+n}(\beta^*) = o_p \left( \frac{1}{\sqrt{N+n}} \right).$

$\square$

# D. Proof of Corollary 4.3

The first series of inequalities is visualized through the variance decomposition in Theorem 4.2, so we omit the proof here. We now consider the prediction-powered inference (PPI++) estimator with power tuning, as proposed by Angelopoulos et al. (2023b), under a design-based setting where the labeling probability $\pi$ is controlled by the experimenter (Zrnic & Candès, 2024a; Egami et al., 2023).

Let $f(X, Z)$ be a pre-trained black-box predictor of $Y$, using both structured and unstructured covariates. To align the comparison with our LASS framework, we define the predicted outcome $Y^* := f(X, Z)$, where the same LLM model is used to generate $Y^*$ and $f(X, Z)$. The PPI++ estimator $\hat{\beta}_{\text{PPI++}}$ solves the estimating equation:

$$\frac{1}{N+n} \sum_{i=1}^{N+n} X_i \left[ \frac{S_i}{\pi} Y_i + \lambda Y_i^* \left( 1 - \frac{S_i}{\pi} \right) - g(X_i^\top \beta) \right] = 0,$$

where $\lambda \in \mathbb{R}$ is a power-tuning parameter balancing observed and predicted responses. By standard arguments in Section A, the asymptotic distribution satisfies

$$\sqrt{N+n}(\hat{\beta}_{\text{PPI++}} - \beta^*) \xrightarrow{d} \mathcal{N}(0, A^{-1} B_{\text{PPI++}} A^{-1}),$$

where

$$B_{\text{PPI++}} = \mathbb{E} \left\{ \left[ \frac{1}{\pi} Y^2 + \lambda^2 (Y^*)^2 \left( \frac{1}{\pi} - 1 \right) - 2\lambda \left( \frac{1}{\pi} - 1 \right) Y Y^* - g^2(X^\top \beta) \right] X X^\top \right\}$$

$$= \mathbb{E} \left\{ \left[ \left( Y - g(X^\top \beta) \right)^2 + (Y - \lambda Y^*)^2 \left( \frac{1}{\pi} - 1 \right) \right] X X^\top \right\}.$$

The asymptotic variance $B_{\text{PPI++}}$ is minimized in $\lambda$ when $\lambda^* = \frac{\mathbb{E}[YY^*]}{\mathbb{E}[(Y^*)^2]}$, yielding the optimal variance:

$$B_{\text{PPI++}}^* = \mathbb{E} \left\{ \left[ \left( Y - g(X^\top \beta) \right)^2 + \left( \frac{1}{\pi} - 1 \right) Y^2 \left( 1 - \frac{\mathbb{E}^2[YY^*]}{\mathbb{E}[Y^2] \mathbb{E}[(Y^*)^2]} \right) \right] X X^\top \right\}.$$

Note that for any fixed $\lambda$, we can bound:

$$\mathbb{E}\left[(Y - \lambda Y^*)^2 XX^\top\right] = \mathbb{E}\left[XX^\top \mathbb{E}\left[(Y - \lambda Y^*)^2 \mid X\right]\right]$$
$$\geq \mathbb{E}\left[XX^\top \operatorname{Var}(Y - \lambda Y^* \mid X)\right].$$

Therefore,

$$B_{\text{PPI++}}^* \geq \mathbb{E}\left\{\left[\left(Y - g(X^\top \beta)\right)^2 + \left(\frac{1}{\pi} - 1\right)\operatorname{Var}(Y - \lambda^* Y^* \mid X)\right] XX^\top\right\}$$
$$\geq \mathbb{E}\left\{\left[\left(Y - g(X^\top \beta)\right)^2 + \left(\frac{1}{\pi} - 1\right)\operatorname{Var}(Y - \omega^*(X)Y^* \mid X)\right] XX^\top\right\}$$
$$= B_{\text{LASS,xz}},$$

where the second inequality holds because $\omega^*(X)Y^*$ is the optimal linear predictor of $Y$ given $Y^*$, i.e., it minimizes the conditional mean squared error. That proves Corollary 4.3. A direct comparison between $\mathbb{V}_{\text{PPI++}}$ and $\mathbb{V}_{\text{SS}}$ is not feasible. Although $Y^*$ may encode rich information from unstructured data, its usefulness depends critically on calibration. In some settings, a poorly calibrated $Y^*$ may perform worse than structured-only machine learning estimators, whereas in others, it may substantially improve efficiency.

# E. Proof of Theorem 4.4

To achieve maximal efficiency, we consider a fine-grained version of the prediction-invariance region $\mathcal{A} \subseteq \mathcal{X}$, defined such that

$$\mathbb{E}\left[\omega_{xzu}^*(X)\,Y_{xzu}^* \mid X \in \mathcal{A}\right] = \mathbb{E}\left[\omega_{xzu}^*(X)\,Y_{xz}^* \mid X \in \mathcal{A}\right].$$

Unlike the coarsened version of $\mathcal{A}$ considered in the main text, here we allow $\omega_{xzu}^*(X)$ to vary freely within $\mathcal{A}$, i.e., even within leaves.

## E.1. Asymptotic Properties with Oracle Region $\mathcal{A}$

We first establish the asymptotic properties of the estimator $\hat{\beta}_{\text{LASS,xzu}}$ assuming that the prediction-invariance region $\mathcal{A}$ is correctly specified. In this subsection, $\mathcal{A}$ is treated as a fixed oracle region and is not assumed to be the limit of any estimated sequence. Let $\hat{\beta}_{\text{LASS,xzu}}(\mathcal{A})$ denote this oracle estimator, to distinguish it from the data-driven version in the main text.

For notational clarity, define the augmented imputation as

$$\mu_{\eta,\mathcal{A}}(O) = m(X) + \mathbf{1}\{X \in \mathcal{A}\}\omega_{xzu}(X)\left\{SY_{xzu}^* + (1 - S)Y_{xz}^* - m_{xz}^*(X)\right\}$$
$$+ \mathbf{1}\{X \notin \mathcal{A}\}\omega_{xz}(X)\left\{Y_{xz}^* - m_{xz}^*(X)\right\},$$

and the pseudo label as

$$\widetilde{Y}_{\eta,\mathcal{A}} = \mu_{\eta,\mathcal{A}}(O)\left(1 - \frac{S}{\pi}\right) + \frac{SY}{\pi}.$$

Then the estimating function can be written compactly as

$$\varphi(O; \beta, \eta, \pi, \mathcal{A}) = X\left\{\widetilde{Y}_{\eta,\mathcal{A}} - g(X^\top \beta)\right\},$$

where $O = (X, Y, Y_{xz}^*, Y_{xzu}^*, S)$, $\eta = (m, m_{xz}^*, \omega_{xz}, \omega_{xzu})$, and $\pi$ is the labeling probability. The estimator with oracle region $\mathcal{A}$ solves

$$\hat{\Phi}_{N+n}(\beta) := \frac{1}{N+n}\sum_{i=1}^{N+n}\varphi(O_i; \beta, \hat{\eta}, \hat{\pi}, \mathcal{A}) = 0.$$

We first discuss the effect of estimating the nuisance functions. Let $\hat{\eta} = (\hat{m}, \hat{m}_{xz}^*, \hat{\omega}_{xz}, \hat{\omega}_{xzu})$ be cross-fitted nuisance estimators. Under boundedness of $X$, the pseudo-labels, and the calibration weights, and under

$$\|\hat{m} - m\|_{L_2(P_X)} + \|\hat{m}_{xz}^* - m_{xz}^*\|_{L_2(P_X)} + \|\hat{\omega}_{xz} - \omega_{xz}\|_{L_2(P_X)} + \|\hat{\omega}_{xzu} - \omega_{xzu}\|_{L_2(P_X)} = o_p(1),$$

the plug-in nuisance effect is asymptotically negligible:

$$\sqrt{N+n}\,\|\mathbb{P}_{N+n}\left[\varphi(O;\beta^*,\hat{\eta},\pi,\mathcal{A})-\varphi(O;\beta^*,\eta,\pi,\mathcal{A})\right]\|=o_p(1).$$

The key reason is that the nuisance errors enter through centered terms. For example, the estimation error of $m_{xz}^*$ contributes terms of the form

$$\mathbb{P}_{N+n}\left[X\left(1-\frac{S}{\pi}\right)\hat{\omega}(X)\{\hat{m}_{xz}^*(X)-m_{xz}^*(X)\}\right],$$

which is mean zero conditional on the nuisance-training sample because the same centering function $m_{xz}^*(X)$ is used in the labeled and unlabeled parts. Cross-fitting and $L_2$ convergence then imply that this term is $o_p((N+n)^{-1/2})$, similar to the proof of Lemma C.4. The same argument applies to the remaining nuisance components. Therefore,

$$\sqrt{N+n}\left\{\hat{\beta}_{\texttt{LASS},\texttt{xzu}}(\mathcal{A};\hat{\eta})-\hat{\beta}_{\texttt{LASS},\texttt{xzu}}(\mathcal{A};\eta)\right\}=o_p(1).$$

To establish consistency of $\hat{\beta}_{\texttt{LASS},\texttt{xzu}}(\mathcal{A})$, it suffices to show that

$$\mathbb{E}\{\varphi(O;\beta,\eta,\pi,\mathcal{A})\}=\mathbb{E}\left[X\{Y-g(X^\top\beta)\}\right],$$

which follows from the definition of the prediction-invariance region $\mathcal{A}$ and the construction of the calibrated pseudo-outcome. Once this holds, consistency follows by an argument analogous to Section C.

We now derive the asymptotic variance for the oracle estimator with population nuisance functions. By the same argument as in Section C,

$$B_{\texttt{xzu}}=\mathbb{V}\mathrm{ar}\left(\sqrt{N+n}\,\hat{\Phi}_{N+n}(\beta^*)\right)=V_1+V_2+2V_{12},$$

where

$$\begin{aligned}
V_1 &= \mathbb{E}\left[\{Y-g(X^\top\beta^*)\}^2 XX^\top\right] \\
&\quad + \mathbb{E}\left[\{Y-\omega_{xz}(X)Y_{xz}^*\}^2\left(\frac{1}{\pi}-1\right)XX^\top\,\Big|\,X\notin\mathcal{A}\right]\mathbb{P}(X\notin\mathcal{A}) \\
&\quad + \mathbb{E}\left[\left\{Y^2-2\omega_{xzu}(X)YY_{xzu}^*+\omega_{xzu}^2(X)\left[(1-\pi)(Y_{xzu}^*)^2+\pi(Y_{xz}^*)^2\right]\right\}\left(\frac{1}{\pi}-1\right)XX^\top\,\Big|\,X\in\mathcal{A}\right]\mathbb{P}(X\in\mathcal{A}),
\end{aligned}$$

$$\begin{aligned}
V_2 &= \mathbb{E}\left[\{m(X)-\omega_{xz}(X)m_{xz}^*(X)\}^2\left(\frac{1}{\pi}-1\right)XX^\top\,\Big|\,X\notin\mathcal{A}\right]\mathbb{P}(X\notin\mathcal{A}) \\
&\quad + \mathbb{E}\left[\{m(X)-\omega_{xzu}(X)m_{xz}^*(X)\}^2\left(\frac{1}{\pi}-1\right)XX^\top\,\Big|\,X\in\mathcal{A}\right]\mathbb{P}(X\in\mathcal{A}),
\end{aligned}$$

and

$$V_{12}=-V_2.$$

The term $V_1$ collects the variance from the observed outcome and pseudo-label components, while $V_2$ and $V_{12}$ arise from centering by the conditional means. Combining these terms gives

$$B_{\texttt{xzu}}=V_1-V_2.$$

If we choose

$$\omega_{xz}^*(X)=\frac{\mathrm{Cov}(Y,Y_{xz}^*\mid X)}{\mathrm{Var}(Y_{xz}^*\mid X)},\qquad \omega_{xzu}^*(X)=\frac{\mathrm{Cov}(Y,Y_{xzu}^*\mid X)}{(1-\pi)\,\mathrm{Var}(Y_{xzu}^*\mid X)+\pi\,\mathrm{Var}(Y_{xz}^*\mid X)},$$

then $B_{\texttt{xzu}}$ is minimized with respect to $\omega(\cdot)$. The minimized asymptotic variance can be written as

$$\begin{aligned}
B_{\texttt{xzu}} &= \mathbb{E}\left[\frac{1}{\pi}\{Y-g(X^\top\beta^*)\}^2 XX^\top\right] \\
&\quad - \mathbb{E}\left[\left(\frac{1}{\pi}-1\right)\{m(X)-g(X^\top\beta^*)\}^2 XX^\top\right] \\
&\quad - \mathbb{E}\left[\left(\frac{1}{\pi}-1\right)\frac{\mathrm{Cov}^2(Y,Y_{xz}^*\mid X)}{\mathrm{Var}(Y_{xz}^*\mid X)}XX^\top\,\Big|\,X\notin\mathcal{A}\right]\mathbb{P}(X\notin\mathcal{A}) \\
&\quad - \mathbb{E}\left[\left(\frac{1}{\pi}-1\right)\frac{\mathrm{Cov}^2(Y,Y_{xzu}^*\mid X)}{(1-\pi)\,\mathrm{Var}(Y_{xzu}^*\mid X)+\pi\,\mathrm{Var}(Y_{xz}^*\mid X)}XX^\top\,\Big|\,X\in\mathcal{A}\right]\mathbb{P}(X\in\mathcal{A}).
\end{aligned}$$

Finally, applying the same M-estimation expansion as in Section C and using Slutsky's theorem, we obtain

$$\sqrt{N+n}\left(\hat{\beta}_{\mathrm{LASS,xzu}}(\mathcal{A}) - \beta^*\right) \xrightarrow{d} \mathcal{N}\left(0, A^{-1}B_{\mathrm{xzu}}A^{-1}\right),$$

where

$$A = \mathbb{E}\left[g'(X^\top\beta^*)XX^\top\right].$$

### E.2. Valid Inference with Estimated $\hat{\mathcal{A}}$

We now return to the case where the prediction-invariance region $\mathcal{A}$ is estimated using the *PIR Selection Algorithm*. Specifically, we refine the test statistic used in Algorithm 2 to accommodate the fine-grained region $\mathcal{A}$ by defining:

$$\hat{\theta}_k = \frac{1}{|I_{2,k}|}\sum_{i\in I_{2,k}} \hat{\omega}^*_{xzu}(X_i)\left(Y^*_{xzu,i} - Y^*_{xz,i}\right), \quad T_k = \frac{\hat{\theta}_k}{\widehat{\mathrm{se}}(\hat{\theta}_k)},$$

where $\hat{\omega}^*_{xzu}(X_i)$ is estimated via cross-fitting as discussed in Section C, and $\widehat{\mathrm{se}}(\hat{\theta}_k)$ is the standard error estimated using cross-fitted bootstrap. Since the CART partition is data-dependent, cross-fitting alone does not remove the first-order randomness of the selected region. To obtain a single unconditional limiting distribution, we assume that the candidate CART partition is asymptotically stable. Let $\{\ell_k\}_{k=1}^K$ denote the limiting finite candidate partition. Define

$$\mathcal{K}_{\mathrm{inv}} = \{k \in \{1,\dots,K\} : \mathbb{E}\left[\omega^*_{xzu}(X)Y^*_{xzu} \mid X \in \ell_k\right] = \mathbb{E}\left[\omega^*_{xzu}(X)Y^*_{xz} \mid X \in \ell_k\right]\},$$

and define the oracle prediction-invariance region as $\mathcal{A} = \bigcup_{k\in\mathcal{K}_{\mathrm{inv}}} \ell_k$. Similarly, let $\hat{\mathcal{K}}_{\mathrm{inv}}$ denote the set of leaves selected by the empirical tests, and define $\hat{\mathcal{A}} = \bigcup_{k\in\hat{\mathcal{K}}_{\mathrm{inv}}} \ell_k$. To ensure valid inference with the estimated prediction-invariance region $\hat{\mathcal{A}}$, we impose the following conditions.

**Assumption E.1** (Validity of Region Identification)**.**

1. **Stability of candidate partition.** There exists a fixed finite partition $\{\ell_k\}_{k=1}^K$ such that the candidate CART partition used by the PIR Selection Algorithm converges to this partition with probability tending to one, up to $P_X$-null sets. In particular, $K$ is fixed and finite, and $\mathbb{P}(X \in \ell_k) > 0$ for all $k$.

2. **Signal separation.** For all $k \notin \mathcal{K}_{\mathrm{inv}}$, the expected difference between augmented and unaugmented predictions is bounded away from zero:
$$\left|\mathbb{E}\left[\omega^*_{xzu}(X)\left(Y^*_{xzu} - Y^*_{xz}\right) \mid X \in \ell_k\right]\right| > c,$$
where $c > 0$ is a fixed constant that does not shrink with sample size.

3. **Vanishing significance level.** The significance level $\alpha$ for multiple testing satisfies $\alpha \to 0$ as $N + n \to \infty$.

Assumption E.1.1 ensures that the data-dependent CART partition does not induce a mixture limit. Assumption E.1.2 ensures that true violations of the prediction invariance condition are detectable with sufficient power. Assumption E.1.3 controls Type I error in large-sample settings. It is worth noting that Assumption E.1.3 is not strictly required for the consistency of $\hat{\beta}_{\mathrm{LASS,xzu}}$; rather, it is needed to recover the same first-order distribution as the oracle estimator. If the significance level $\alpha$ remains fixed (e.g., at 5%), the estimated region $\hat{\mathcal{A}}$ may fail to contain invariant subregions with non-vanishing probability. As a result, the asymptotic distribution of $\hat{\beta}_{\mathrm{LASS,xzu}}$ becomes a mixture distribution.

**Lemma E.2.** *Under Assumption 4.1, Assumptions C.1-C.3 and Assumption E.1, the feasible estimator using the estimated region $\hat{\mathcal{A}}$ is asymptotically equivalent to the oracle estimator:*

$$\hat{\beta}_{LASS,xzu} = \hat{\beta}_{LASS,xzu}(\mathcal{A}) + o_p\left(\frac{1}{\sqrt{N+n}}\right).$$

Combining Lemma E.2 with the results from Section E.1, we conclude that the asymptotic properties established for the oracle estimator $\hat{\beta}_{\mathrm{LASS,xzu}}(\mathcal{A})$ carry over to the feasible estimator $\hat{\beta}_{\mathrm{LASS,xzu}}$. This completes the proof of Theorem 4.4.

*Proof of Lemma E.2.* We sketch the proof by showing that the remainder term

$$R_{N+n} := \frac{1}{N+n} \sum_{i=1}^{N+n} \left\{ \varphi\left(O_i; \beta, \hat{\eta}, \hat{\pi}, \hat{\mathcal{A}}\right) - \varphi\left(O_i; \beta, \hat{\eta}, \hat{\pi}, \mathcal{A}\right) \right\}$$

is asymptotically negligible. Using the definition of $\varphi$, the difference reduces to:

$$R_{N+n} = \frac{1}{N+n} \sum_{i=1}^{N+n} X_i \bigg\{ \left[ \mathbf{1}\{X_i \in \hat{\mathcal{A}} \setminus \mathcal{A}\} - \mathbf{1}\{X_i \in \mathcal{A} \setminus \hat{\mathcal{A}}\} \right]$$
$$\hat{\omega}_{xzu}^*(X_i) \left( S_i Y_{xzu,i}^* + (1-S_i) Y_{xz,i}^* - \hat{m}_{xz}^*(X_i) \right)$$
$$+ \left[ \mathbf{1}\{X_i \in \mathcal{A} \setminus \hat{\mathcal{A}}\} - \mathbf{1}\{X_i \in \hat{\mathcal{A}} \setminus \mathcal{A}\} \right]$$
$$\hat{\omega}_{xz}^*(X_i) \left( Y_{xz,i}^* - \hat{m}_{xz}^*(X_i) \right) \bigg\} \left( 1 - \frac{S_i}{\hat{\pi}} \right).$$

By Assumption E.1.1, with probability tending to one, the candidate CART partition coincides with the fixed finite partition $\{\ell_k\}_{k=1}^K$. Hence, it suffices to analyze the leaf-level testing errors on this stable partition.

For $k \notin \mathcal{K}_{\text{inv}}$, Assumption E.1.2 gives a non-zero signal bounded below by a universal constant $c > 0$. By Assumptions 4.1 and E.1.2, the standard error of the test statistic satisfies $\text{se}(\hat{\theta}_k) = o_p(1)$. Therefore, the signal-to-noise ratio diverges. Under the boundedness conditions in Assumption 4.1, standard concentration inequalities imply that the Type II error decays exponentially fast. Hence,

$$\frac{1}{N+n} \sum_{i=1}^{N+n} \mathbf{1}\{X_i \in \hat{\mathcal{A}} \setminus \mathcal{A}\} = o_p\left( \frac{1}{\sqrt{N+n}} \right).$$

Similarly, by Assumption E.1.3 on the vanishing significance level, we control the false rejection of invariant leaves. Since $K$ is fixed and $\alpha \to 0$,

$$\mathbb{P}\left( \mathcal{K}_{\text{inv}} \not\subseteq \hat{\mathcal{K}}_{\text{inv}} \right) = o(1).$$

Consequently,

$$\frac{1}{N+n} \sum_{i=1}^{N+n} \mathbf{1}\{X_i \in \mathcal{A} \setminus \hat{\mathcal{A}}\} = o_p(1).$$

Next, we bound the magnitude of the remaining summand terms. For each $\ell_k \subset \mathcal{A}$, by CLT we have:

$$\frac{1}{|\ell_k|} \sum_{i \in \ell_k} X_i \hat{\omega}_{xzu}^*(X_i) \left( S_i Y_{xzu,i}^* + (1-S_i) Y_{xz,i}^* - \hat{m}_{xz}^*(X_i) \right) \left( 1 - \frac{S_i}{\hat{\pi}} \right) = O_p\left( \frac{1}{\sqrt{N+n}} \right),$$
$$\frac{1}{|\ell_k|} \sum_{i \in \ell_k} X_i \hat{\omega}_{xz}^*(X_i) \left( Y_{xz,i}^* - \hat{m}_{xz}^*(X_i) \right) \left( 1 - \frac{S_i}{\hat{\pi}} \right) = O_p\left( \frac{1}{\sqrt{N+n}} \right).$$

The first disagreement set, $\hat{\mathcal{A}} \setminus \mathcal{A}$, has exponentially small probability because non-invariant leaves are rejected with exponentially high probability. The second disagreement set, $\mathcal{A} \setminus \hat{\mathcal{A}}$, may occur with probability $o(1)$, but its contribution is multiplied by the centered leaf-level averages displayed above, which are $O_p((N+n)^{-1/2})$. Hence overall remainder satisfies:

$$R_{N+n} = o_p\left( \frac{1}{\sqrt{N+n}} \right).$$

$\square$

## F. Extension: LASS under covariate shift

In this section, we extend the LASS estimator introduced in the main text to accommodate covariate shift. Under covariate shift, the target parameter remains defined on the overall population $\beta^* : \mathbb{E}\left[ X \left( Y - g\left( X^\top \beta \right) \right) \right] = 0$. However, when labeled data are drawn from a shifted distribution, naively fitting a GLM using only the labeled subset will generally yield

inconsistent estimates of $\beta^*$. When the covariate shift occurs only over the structured covariates of interest $X$ (or a subset thereof), we can estimate the labeling mechanism $\pi(x) = \mathbb{P}(S = 1 \mid X = x)$ using standard machine learning methods. We then construct the following doubly robust pseudo-label:

$$\tilde{Y}_{\mathrm{DR},i} = \frac{S_i Y_i}{\hat{\pi}(X_i)} + [\hat{m}(X_i) + \hat{\omega}^*(X_i)(Y_i^* - \hat{m}^*(X_i))]\left(1 - \frac{S_i}{\hat{\pi}(X_i)}\right),$$

and solve for $\hat{\beta}_{\mathrm{LASS,xz}}$ using the estimating equation:

$$\frac{1}{N+n}\sum_{i=1}^{N+n}\left\{\tilde{Y}_{\mathrm{DR},i} - g\left(X_i^\top \beta\right)\right\} X_i = 0.$$

To ensure valid statistical inference, in addition to Assumption 1 and Assumptions A.1–A.3, the nuisance estimators $\hat{\pi}(x)$, $\hat{m}(x)$, $\hat{m}^*(x)$, and $\hat{\omega}^*(x)$ must converge at a rate faster than $o_p(n^{-1/4})$ (Chernozhukov et al., 2018). Such rates are typically attainable when the underlying functions are sufficiently smooth or sparse.

In cases where covariates are misaligned across labeled and unlabeled data sources, we can nonparametrically identify an invariance region $\mathcal{R} \subseteq \mathcal{X}$ such that:

$$\mathbb{E}[Y_{xzu} \mid X = x] = \mathbb{E}[Y_{xz} \mid X = x], \quad \text{for all } x \in \mathcal{R}.$$

Within this region, we can selectively incorporate additional covariates following the selective integration strategy discussed in Section 3.2.

Finally, if there is a shift in unstructured covariates that cannot be aligned across data sources, consistent estimation of $\beta^*$ at the root-$n$ rate becomes impossible. We omit technical details in this case for brevity.

## G. Additional Comparison with Deep-Learning-Based Pseudo-Labels

This section compares the proposed LASS framework with a natural deep-learning-based alternative. The comparison focuses on the label-scarce setting, where the labeled sample is limited but a large unlabeled sample is available. In LASS, the pseudo-label $Y_{xz}^*$ is generated by a pretrained LLM using $(X, Z)$ and is then calibrated using the labeled data. In contrast, a DL-based alternative first trains a task-specific prediction model on the labeled sample and then uses its predictions as pseudo-labels. The downstream calibration and semi-supervised estimating equation are otherwise the same.

### G.1. Theoretical comparison

The main distinction is whether the pseudo-label model must be learned from the labeled sample. In LASS, the LLM is pretrained, so the labeled data are used only to calibrate the pseudo-labels and estimate the target parameter. The calibration step estimates quantities such as $m(x) = \mathbb{E}(Y \mid X = x)$, $m_{xz}^*(x) = \mathbb{E}(Y_{xz}^* \mid X = x)$, and $\omega_{xz}^*(x) = \mathrm{Cov}(Y_{xz}^*, Y \mid X = x)/\mathrm{Var}(Y_{xz}^* \mid X = x)$.

For a DL-based alternative, the pseudo-label model itself must first be trained using the labeled sample. Let $\hat{\mu}_{\mathrm{DL}}(X, Z)$ denote the resulting DL predictor, and suppose its prediction error is of order $\|\hat{\mu}_{\mathrm{DL}} - \mu_{\mathrm{DL}}^*\|_{L_2(P)} = O_p(n^{-\alpha})$ for some $\alpha > 0$. Suppose also that the calibration nuisance functions are estimated at rate $\max\{\|\hat{m} - m\|_{L_2(P_X)}, \|\hat{m}^* - m^*\|_{L_2(P_X)}, \|\hat{\omega} - \omega\|_{L_2(P_X)}\} = O_p(n^{-\theta})$ for some $\theta > 0$. Then the DL-based estimator is affected by both the error from training the DL predictor and the error from calibration, leading to the schematic bias order $|\mathrm{Bias}(\hat{\beta}_{\mathrm{DL},xz})| = O_p(n^{-\alpha}) + O_p(n^{-\theta})$.

By contrast, the LASS estimator does not need to train the pseudo-label model from the labeled sample. Its finite-sample bias is mainly driven by calibration error, with schematic order $|\mathrm{Bias}(\hat{\beta}_{\mathrm{LASS},xz})| = O_p(n^{-\theta})$. Therefore, when the DL predictor is difficult to train from limited labeled data, so that $0 < \alpha < \theta$, the additional term $O_p(n^{-\alpha})$ can dominate the finite-sample bias of the DL-based estimator.

The variance comparison depends on how informative the pseudo-label is for the true outcome. By Theorem 4.2, the efficiency gain from using the pseudo-label is driven by $\mathrm{Cov}^2(Y_{xz}^*, Y \mid X)/\mathrm{Var}(Y_{xz}^* \mid X)$. Thus, a more predictive pseudo-label leads to larger variance reduction after calibration. A DL-based pseudo-label can also be useful if it is strongly predictive of $Y$. However, in small labeled samples, a task-specific DL model may be less stable because it must learn the

*Table 8.* Standard deviation comparison between DL-based pseudo-labels and LASS. The DL-based method fine-tunes DistilBERT on the labeled sample, while LASS uses pretrained LLM pseudo-labels.

| Covariate | DL ($n = 100$) | DL ($n = 300$) | DL ($n = 500$) | LASS ($n = 100$) |
|---|---|---|---|---|
| Lexical diversity | 0.65 | 0.60 | 0.54 | 0.52 |
| Race | 0.64 | 0.61 | 0.59 | 0.58 |
| Sex | 0.73 | 0.70 | 0.67 | 0.65 |
| Pronoun ratio | 0.68 | 0.63 | 0.58 | 0.56 |
| Filler words ratio | 0.52 | 0.48 | 0.45 | 0.43 |
| Verb ratio | 0.49 | 0.44 | 0.40 | 0.38 |

*Table 9.* Absolute bias comparison between DL-based pseudo-labels and LASS.

| Covariate | DL ($n = 100$) | DL ($n = 300$) | DL ($n = 500$) | LASS ($n = 100$) |
|---|---|---|---|---|
| Lexical diversity | 0.079 | 0.058 | 0.037 | 0.026 |
| Race | 0.062 | 0.046 | 0.031 | 0.024 |
| Sex | 0.054 | 0.040 | 0.027 | 0.021 |
| Pronoun ratio | 0.071 | 0.053 | 0.034 | 0.025 |
| Filler words ratio | 0.049 | 0.036 | 0.023 | 0.018 |
| Verb ratio | 0.044 | 0.032 | 0.021 | 0.017 |

prediction rule from limited data. A pretrained LLM can use external pretraining information and may therefore provide a stronger pseudo-label source in this setting (Howard & Ruder, 2018).

A similar issue arises under covariate misalignment. The prediction-invariance-region procedure compares predictions based on $(X, Z)$ and $(X, Z, U)$. If the pseudo-labels are generated by a DL model trained only on the small labeled sample, the comparison may be unstable because the DL predictions themselves have substantial estimation error. In contrast, LASS uses pretrained LLM predictions and therefore avoids training the pseudo-label model from the limited labeled sample, which can make the identification of the prediction-invariance region more stable.

### G.2. Simulation comparison

We further conduct a simulation study comparing LASS with a DL-based variant. For the DL-based method, we fine-tune DistilBERT as a text classifier using only the labeled sample and use its predicted labels as pseudo-outcomes. The remaining steps are identical to LASS: both methods use calibrated pseudo-labels and the same semi-supervised estimating equation. Thus, the comparison isolates the effect of the pseudo-label source: the DL-based method uses predictions from fine-tuned DistilBERT, whereas LASS uses predictions from a pretrained LLM.

For the DL-based method, we consider labeled sample sizes $n = 100, 300, 500$, with unlabeled sample size fixed at $N = 1000$. For LASS, we report the result with only $n = 100$ labeled samples. Table 8 reports the empirical standard deviation, and Table 9 reports the absolute bias.

The results are consistent with the discussion above. In Table 8, the standard deviation of the DL-based method decreases as the labeled sample size increases. However, LASS with only $n = 100$ labeled samples still achieves smaller standard deviation than the DL-based method across all covariates. Table 9 shows the same pattern for absolute bias. Although the bias of the DL-based method decreases as $n$ increases from 100 to 500, LASS with only $n = 100$ still yields smaller finite-sample bias across all covariates.

Overall, these results suggest that, under label scarcity, pretrained LLM pseudo-labels can provide a more effective surrogate for semi-supervised inference than pseudo-labels generated by a task-specific DL model trained directly on the limited labeled sample. This leads to both lower variance and smaller finite-sample bias, consistent with the efficiency pattern observed for LASS in the main empirical studies.

