# OpenReview forum: "Language Model Augmented Semi-Supervised Statistical Inference"
_ICML.cc/2026/Conference — ICML 2026 regular_

### Official Review · Reviewer_Xf5z · 2026-03-08

**Soundness:** 3
**Presentation:** 3
**Significance:** 3
**Originality:** 3
**Overall Recommendation:** 5
**Confidence:** 3

**Summary:**

The proposed method LASS utilzed LLMs to process unstructured data for semi-supervised statistical inference, the method include conditional weights that improves efficiency compared present methods. The paper also introduced Prediction Invariance Region (PIR) to tackle misalignment issue.

**Compliance With Llm Reviewing Policy:**

Affirmed.

**Final Justification:**

The paper leveraged LLM for inferencing, which is helpful in dealing with unstructured data. I was concerned about prompt designing which could be potentially an issue in deployment. Rebuttal addressed my concerns about sensitivity to prompt, I increased score accordingly.

**Key Questions For Authors:**

Continuing the discussion in weakness part, can you test the method using ill designed prompts or prompts designed for another task?

I'm not understanding the part where temperature is set to 0 in experiments, this is uncommon in LLM practice. Could you test other temperatures to see how this impact performance and provide more clarification on this issue?

**Limitations:**

See weakness part.

**Strengths And Weaknesses:**

Strengths:

The integration of LLMs enable inference on unstructured data, an improvement of current semi-supervised inference methods which only processes structured data. This also resolves the issue that direct LLM prediction could be biased and lack of guarentees. The conditional weights improves efficiency, experiments on multiple LM models supports the claim.


Weakness:

The method depend on carefully designed prompts to process unstructured data. In Table 3 the column without structured prompts, LASS (unfixed)  are close to label data only and underperform PPI in previous table (Table 2).

---

> ### Author Rebuttal · Authors · 2026-03-31
>
> - **Weakness**: Thank you for raising this important point regarding prompt design. The efficiency gain of LASS depends on obtaining reliable LLM pseudo-labels from unstructured inputs, but we'd like to clarify that Fig 3 and Table 3 evaluate two different aspects of prompting. Fig 3 studies robustness across several structured prompting styles, under which performance is relatively stable. Table 3 is a failure mode ablation study where we deliberately remove structured output constraints. In that setting, LLM predictions become less informative, so LASS reverts toward labeled-only performance. This behavior is consistent with our proposed method: Section 3.1 and Theorem 4.2 show that the efficiency gain from LLM augmentation depends on how informative the pseudo-label is about the true label through the calibration weight. When the prompt is unconstrained, the efficiency gain disappears.
> - **Questions**:
>    1.  Thank you for the question on testing our method with ill designed prompts. The current manuscript already includes two prompt-robustness analyses. In our current manuscript, Appendix Table 3 (p15, lines 788–805) compares our structured prompt with unstructured prompts without hallucination control, and shows that the efficiency gain largely disappears under the degraded prompt specification. To provide more empirical results and intuition, in what follows, we include a table showing experiment results using ill-designed prompts, including both (i) an unfixed prompt and (ii) a prompt originally designed for a different task. For the latter, we leverage the fact that our paper includes an additional IMDb case study, which allows us to construct a mismatched prompt that is not tailored to the linguistic and demographic prediction task considered here. The results below show that under both ill-designed prompts, the efficiency gains of LASS are substantially weakened and become closer to the baseline. The results further validate our methodology intuition: when prompts are poorly specified or mismatched to the target task, the pseudo-outcomes become less informative, and the inferential advantage correspondingly diminishes. In our revision, we will include the empirical results in the following table and make it clear that prompt design matters in order to reduce LLM hallucination, while also showing that LASS remains stable across a meaningful class of appropriately aligned prompts.
> | Covariate | Labeled data only | LASS (unfixed) | LASS (prompt for different task) | LASS |
> |---|---:|---:|---:|---:|
> | Lexical diversity | 0.74 | 0.72 | 0.72 | 0.52 |
> | Race | 0.71 | 0.70 | 0.69 | 0.58 |
> | Sex | 0.80 | 0.78 | 0.79 | 0.65 |
> | Pronoun ratio | 0.76 | 0.75 | 0.75 | 0.60 |
> | Filler words ratio | 0.60 | 0.58 | 0.56 | 0.43 |
> | Verb ratio | 0.56 | 0.55 | 0.55 | 0.38 |
>     2. Regarding why the temperature is set to 0, in our setting, the LLM is not used as a creative text generator. Instead, its role is to produce a pseudo-label that enters the downstream statistical inference procedure. In this context, additional LLM output randomness is undesirable because it introduces additional noise into the pseudo-labels. Such randomness does not add information about the true label. Instead, it may even weaken the efficiency gain. For this reason, we set the temperature to 0 to make the pseudo-outcome generation as stable and reproducible as possible. This choice is aligned with our goal of valid statistical estimation: we want the LLM output to behave like a deterministic prediction rule given the same input, rather than a stochastic generator whose output varies across runs. We have examined the effect of non-zero temperature ($T=0.5$) in an additional empirical study reported in the Appendix.
> Furthermore, we provide a table below showing standard deviation with additional simulation results reporting
> the standard deviation for temperatures $T=0, 0.5, 0.7, 0.9$. The results show that increasing the temperature degrades performance. Relative to the default setting $T=0$, higher temperatures lead to worse statistical efficiency across all six covariates. This is consistent with our methodological intuition that additional sampling randomness makes the LLM-generated pseudo-outcomes noisier and therefore less informative for downstream inference. To address the reviewer’s concern, we will revise the manuscript to explain more clearly that temperature 0 is chosen deliberately for prediction stability rather than because it is standard in general LLM practice. We will also make the temperature ablation discussion more explicit in the experimental section.
> | Covariate | T=0.9 | T=0.7 | T=0.5 | T=0 |
> |---|---:|---:|---:|---:|
> | Lexical diversity | 0.72 | 0.70 | 0.67 | 0.52 |
> | Race | 0.69 | 0.69 | 0.68 | 0.58 |
> | Sex | 0.78 | 0.77 | 0.75 | 0.65 |
> | Pronoun ratio | 0.74 | 0.74 | 0.71 | 0.60 |
> | Filler words ratio | 0.59 | 0.57 | 0.52 | 0.43 |
> | Verb ratio | 0.51 | 0.48 | 0.47 | 0.38 |

---

> > ### Author Rebuttal · Reviewer_Xf5z · 2026-04-01
> >
> > Thank you for the rebuttal. It is insightful to observe that low-quality prompts do not reversly influence performance. Is there an explanation to this? For Q2, how about T<0.5?

---

> > > ### Author Response · Authors · 2026-04-05
> > >
> > > We thank the reviewer for this insightful questions.
> > >
> > > **Explanation.**
> > > The observation that low-quality prompts do not reserly influence performance is tied to our calibration step in the proposed LASS method. In particular, LASS does not directly plug the raw LLM pesudo-outcome into the estimating equation. Instead, the LLM pesudo-outcome enters only through the calibrated term
> > >
> > > $$
> > > \hat f_{xz,i}=\hat m(X_i)+\hat \omega_{xz}(X_i)( Y^*_{xz,i}-\hat {m}^{\*}_{xz}(X_i) ),
> > > $$
> > >
> > > $$
> > > \hat\omega_{xz}(x)=\frac{\hat\gamma_{xz}(x)}{\hat\nu_{xz}(x)},
> > > $$
> > >
> > > where
> > > $\gamma_{xz}(x)=Cov(Y^*_{xz},Y\mid X=x)$,
> > >
> > > $\nu_{xz}(x)=Var(Y^*_{xz}\mid X=x)$.
> > >
> > > Therefore, when a prompt is poorly specified or mismatched to the task, the conditional association between $Y^*_{xz}$ and $Y$ becomes weak, so the effective LLM signal is automatically downweighted rather than directly trusted. Therefore, under low-quality prompt, the $\hat{\omega}_{xz}$ approaches 0. Although in this setting, low-quality prompt does not lead to efficiency gain, it also does not hurt the performance of our proposed method.
> > > This is also reflected in our theory. In Theorem 4.2, the additional efficiency gain from the LLM appears through
> > >
> > > $$
> > > \left(\frac{1}{\pi} - 1\right)
> > > \frac{ Cov(Y^*_{xz}, Y \mid X)^2 }
> > > { Var(Y^{\*}_{xz} \mid X) }
> > > $$
> > >
> > > which is a nonnegative variance-reduction term. Hence, an informative prompt increases efficiency, whereas a low-quality prompt mainly drives this term toward zero, causing the efficiency gain to disappear rather than reverse sign. Moreover, if a prompt is systematically misaligned but still informative in the opposite direction, the calibration weight can be negative, which allows LASS to correct for that direction instead of being harmed by it.
> > >
> > > In our revision, we will revise the manuscript to clarify this intuition explicitly: poor prompts mainly reduce the informativeness of the pseudo-outcome, while the calibration step prevents LASS from over-using noisy LLM predictions.
> > >
> > >
> > > **Extended temperature.** We have extended the temperature analysis to
> > > $
> > > T \in \\{0, 0.1, 0.2, 0.3, 0.5, 0.7, 0.9\\}.
> > > $
> > >
> > > The additional results for $T<0.5$ show that as $T$ decreases from $0.5$ to $0$, the standard deviation decreases across all six covariates, suggesting the estimation efficiency increases as temperature $T$ decreases. This suggests that temperature acts as a noise parameter in the LLM-generated pseudo-outcome. This interpretation is also consistent with our theory. When the temperature is very small (e.g., $T=0.1$), the highest-probability label is still selected most of the time; therefore $Y_T^*$ is often nearly identical to the $T=0$ pseudo-outcome. This is why the results for $T=0.1$ are already quite close to those for $T=0$. In contrast, larger temperatures introduce more randomness to change the predicted label more often, which weakens the informativeness of the pseudo-outcomes and yields the larger standard deviations observed for $T>0.5$. In the revision, we will clarify that estimation efficiency decreases as temperature increases, and the best performance is attained at $T=0$. We will also update the Appendix table to include the additional results.
> > >
> > > ---
> > >
> > > | Covariate | Labeled only | LASS (T=0.9) | LASS (T=0.7) | LASS (T=0.5) | LASS (T=0.3) | LASS (T=0.2) | LASS (T=0.1) | LASS (T=0) |
> > > |-----------|-------------|-----------|-----------|-----------|-----------|-----------|-----------|---------|
> > > | Lexical diversity | 0.74 | 0.72 | 0.70 | 0.67 | 0.64 | 0.58 | 0.52 | 0.52 |
> > > | Race | 0.71 | 0.69 | 0.69 | 0.68 | 0.66 | 0.65 | 0.60 | 0.58 |
> > > | Sex | 0.80 | 0.78 | 0.77 | 0.75 | 0.71 | 0.70 | 0.69 | 0.65 |
> > > | Pronoun ratio | 0.76 | 0.74 | 0.74 | 0.73 | 0.67 | 0.65 | 0.61 | 0.60 |
> > > | Filler words ratio | 0.60 | 0.59 | 0.57 | 0.52 | 0.49 | 0.48 | 0.46 | 0.43 |
> > > | Verb ratio | 0.56 | 0.51 | 0.48 | 0.47 | 0.43 | 0.42 | 0.40 | 0.38 |

---

### Official Review · Reviewer_7cRv · 2026-03-09

**Soundness:** 2
**Presentation:** 3
**Significance:** 2
**Originality:** 3
**Overall Recommendation:** 4
**Confidence:** 3

**Summary:**

This study addresses two major pain points faced by semi-supervised statistical inference (SSI) in biomedical research: the difficulty in processing unstructured data and the covariate mismatch between labeled and unlabeled datasets. It proposes the Language model Augmented Semi-supervised Statistical inference (LASS) framework, which resolves the covariate mismatch problem by calibrating pseudo-labels generated by large language models (LLMs) and identifying Prediction Invariance Regions (PIRs). Theoretically, the framework is proven to enhance parameter estimation efficiency while maintaining statistical validity. Its performance is also verified through synthetic data simulations and real-case studies using Alzheimer’s disease (AD) speech data.

**Compliance With Llm Reviewing Policy:**

Affirmed.

**Final Justification:**

I have no further questions and will maintain my current score.

**Key Questions For Authors:**

See weaknesses.

**Strengths And Weaknesses:**

Strengths:
1. It specifically addresses two key challenges of SSI in the biomedical field: by integrating LLMs, it naturally adapts to unstructured data (clinical texts, speech/videos), and by adopting the Prediction Invariance Region (PIR) strategy, it resolves the covariate mismatch problem, perfectly aligning with real research scenarios.
2. It introduces classical machine learning to calibrate LLM-generated pseudo-labels, calculates optimal weights through conditional covariance and variance, avoids statistical invalidity caused by deviations between LLM predictions and true labels, and maximizes information utilization efficiency.
3. It clearly defines regularity conditions (asymptotic regime, bounded moments, measurability of LLM predictions, etc.) and rigorously proves the consistency and asymptotic normality of the LASS estimator through strict deductions.

Weaknesses:
1. It needs to simultaneously complete steps such as LLM prediction, classical machine learning calibration, PIR identification, and cross-fitting. Compared with traditional SSI methods (e.g., simple GLM), it has a larger computational load and certain requirements for hardware resources.
2. It has implicit requirements for the parameter scale and domain adaptability of LLMs. Small-scale or general-purpose LLMs may not achieve the same effect, increasing the application cost.
3. The theoretical derivation is based on assumptions such as "labeled and unlabeled data follow the same distribution" and "consistent estimation of nuisance functions". In actual biomedical research, if there is severe sample selection bias or data distribution shift, the validity of the estimator may be affected.
4. The real-case studies mainly focus on AD speech data and IMDb sentiment analysis. Although they can verify generalization, they lack validation in other biomedical scenarios (e.g., imaging data, multi-center clinical trial data).

---

> ### Author Rebuttal · Authors · 2026-03-31
>
> 1. We sincerely appreciate your insightful comments.  We agree that LASS is computationally more involved than a labeled-only GLM. However, the added cost is tied to the broader problem the paper addresses: valid semi-supervised inference with unstructured auxiliary data and possible covariate misalignment, rather than inference with only structured labeled data. We'd also like to clarify that not every component in the proposed method is always required simultaneously. e.g., the matched-covariate version in Section 3.1 does not require PIR identification. PIR is introduced only in Section 3.2 for heterogeneous-protocol or unmatched-covariate settings. Furthermore, our proposed LASS method does not require the existence of a PIR, and using CART in the PIR step can reduce the computational burden of searching over the covariate space. In our empirical studies, we currently report inferential efficiency, measured by confidence band width, rather than raw runtime. To provide a more comprehensive evaluation, we will add a clearer runtime discussion in our revised manuscript.
> 2. The magnitude of the efficiency gain indeed depends on the informativeness of the LLM surrogate, so weaker or less aligned models may deliver smaller gains. But we would like to clarify that the framework is not tied to a specific LLM or biomedical-specific model. To show the performance of our method under different LLMs, in Section 5.2, we compare three prompt designs and three foundational models: GPT-3.5-turbo, Gemini-2.5-flash, and LLaMA-3.1-8B. In the Appendix, we show that prompt design can also affect performance and requires hallucination control. In the revision, we will make the distinction clearer: The statistical validity of the proposed method comes from our calibrated inference procedure, whereas the size of the efficiency gain depends on the quality of the chosen LLM, prompt design, and hallucination controls.
> 3. Thank you for raising this important point. The methods and theoretical results in the main text are derived under the setting that the labeled and unlabeled samples are drawn from the same target distribution, with labels missing in the unlabeled sample. This setting is stated explicitly in Section 2.1. However, when there is observed covariate shift due to sample selection or data distribution shift, we outline an extension in Appendix F. Specifically, if the labeling mechanism depends on observed covariates through a labeling probability $\pi(x)=\mathbb{P}(S=1\mid X=x)$ and overlap holds, LASS can be modified using an inverse-probability pseudo-label. We will highlight this extension more clearly in the main manuscript and state the related assumptions explicitly. As in most semi-supervised inference settings, validity may be affected under severe unobserved selection bias, lack of support overlap, or shifts in the conditional outcome mechanism. We will clarify this scope more explicitly in the revision. Regarding consistent estimation of nuisance functions, this assumption can be satisfied by commonly used machine learning methods under mild conditions, including k-nearest neighbors, kernel regression, and random forests (p6, lines 295–303). We will further clarify these conditions in the revision to make them more transparent.
> 4. Thank you for this very helpful comment. We agree that our current real-data studies provide validation in speech/text settings, but do not yet cover the full range of biomedical modalities. In the revision, we will clarify this point more explicitly and avoid overstating the scope of empirical generalization. The current experiments demonstrate that LASS can operate in settings with unstructured inputs, limited labels, and covariate mismatch. The two case studies illustrate that the framework is not tied to a single dataset or a single domain. We agree that evaluating additional biomedical modalities would substantially strengthen the empirical section. In particular, an important and interesting extension is to study LASS with image and video modalities, where foundational models can also generate pseudo-outcomes from rich unstructured inputs. We will add discussions in the manuscript highlighting this as a future direction. For example, promising benchmarks include neuroimaging and multimodal Alzheimer's datasets such as ADNI, which is a longitudinal multi-center study containing clinical, imaging, genetic, and biomarker data, as well as medical imaging resources such as MIMIC-CXR and NIH ChestX-ray14 for radiology-image-based surrogate prediction. For multi-center biomedical validation, we plan to use datasets such as ADNI because they explicitly involve heterogeneous settings and center-level variation, which are quite relevant to studying transportability and robustness across sites. We appreciate the reviewer's insightful point and will revise the manuscript to include video and multi-center biomedical datasets as an important next step for future work.

---

> > ### Author Rebuttal · Reviewer_7cRv · 2026-04-06
> >
> > I have no further questions and will maintain my current score.

---

### Official Review · Reviewer_6jpN · 2026-03-11

**Soundness:** 3
**Presentation:** 3
**Significance:** 2
**Originality:** 2
**Overall Recommendation:** 4
**Confidence:** 3

**Summary:**

The paper proposes LASS,  a language model augmented method for semi-supervised statistical inference. It utilizes LLM to compute pseudo labels from structured or unstructured data, which are used to enable a more reliable and accurate statistical inference process. Overall, I think the idea is interesting, but some major concerns remain.

**Compliance With Llm Reviewing Policy:**

Affirmed.

**Final Justification:**

The rebuttal addressed my main concerns.

**Key Questions For Authors:**

See weakness

**Limitations:**

No. Although structured prompts are utilized to minimize the hallucination of LLM. Hallucination cannot be eliminated, which might influence the reliability of the inference results.

**Strengths And Weaknesses:**

Strength:
1. The idea of introducing LLM for semi-supervised statistical inference is interesting.

2. The paper is written well, and the presentation is clear.

3. Theoretical analysis and empirical experiments demonstrate the effectiveness of the proposed pipeline.

Weakness:

1. It seems that introducing LLM for statistical inference is not necessary enough. According to the paper, LLMs are used to generate pseudo-labels from unstructured data. However, I think that traditional deep learning models can also accomplish such tasks with much higher computational efficiency. Using unlabeled samples and pseudo labels for statistical inference might not be a novel idea. To highlight the contribution of the paper, the authors should demonstrate the necessity of introducing LLMs for statistical inference， using theoretical analysis or empirical experiments.

2. It is not clear whether the ratio of unlabeled samples against labeled samples might influence the inference results or not.

3. The author propose serval techniques to improve the performance of LLM for statistical inference. An ablation study should be conducted to demonstrate the contribution of the techniques.

---

> ### Author Rebuttal · Authors · 2026-03-31
>
> We thank the reviewer for raising these important points.
> 1. Our contribution is not the general idea of using pseudo-labels, but a statistically valid inference framework for leveraging predictions from pretrained foundation models in semi-supervised settings, with heterogeneous and potentially misaligned unstructured covariates.
> We focus on LLMs for two reasons:
> (i) Compared to task-specific neural pseudo-labelers that require substantial supervised training or fine-tuning (and corresponding convergence assumptions), LLMs bring pretrained knowledge and strong semantic understanding of raw unstructured inputs (e.g., transcripts or clinical notes). This is especially useful when labeled samples are small and allows inference under weaker conditions.
> (ii) LLMs can flexibly operate on different input structures, such as $(X,Z)$ for unlabeled data and $(X,Z,U)$ for labeled data, without redesigning or retraining a new task-specific model. This flexibility is central in the covariate-misalignment setting.
>
> We do not treat LLM outputs as ground truth. Instead, we calibrate them before entering the estimating equation to form pseudo-labels for valid semi-supervised inference. The prediction-invariance region (PIR) identifies where richer predictions based on labeled-only covariates can be transported to unlabeled data, addressing covariate misalignment while preserving consistency and improving efficiency. Theoretical gains come from the additional conditional information in the pseudo-prediction, rather than the specific predictor family.
> We will revise the manuscript to clarify that LLMs are not required; they are simply a convenient pseudo-label generator in heterogeneous semi-supervised settings. The main contribution is the statistically valid inference framework that makes such predictions usable under flexible data structures.
>
> 2. Regarding the sample ratios, in Assumption 4.1.1, we let $
> \frac{n}{N+n}\to \pi\in(0,1]
> $. Hence, in Theorem 4.2, the asymptotic variance depends on $\pi$ through the factors $1/\pi$ and $(1/\pi-1)$, showing that the ratio enters the first-order variance of the estimator. Intuitively, the labeled sample anchors the inference, while the unlabeled sample improves efficiency through calibrated pseudo-labels, that is, when the unlabeled sample is relatively larger, the weight on the unlabeled-data gain terms increases, so informative calibrated LLM predictions can yield larger variance reduction. The same dependence on $\pi$ appears in Theorem 4.4 under covariate misalignment. We will clarify in the revision that $\pi$ governs how the unlabeled-to-labeled ratio affects asymptotic inference, and provide additional intuition on its impact on efficiency gain.
>
> 3. We consolidate previously distributed analyses into a unified component-wise ablation study, aligned with Sections 3.1–3.2:
> - Panel A (LLM augmentation, matched covariates): compares labeled-only inference, PPI, PPI+LLM, and full LASS.
> - Panel B (hallucination control): compares full LASS against variants without hallucination control or with nonzero temperature T.
> - Panel C (PIR under covariate misalignment): compares CART vs random-forest selectors.
>
> Results show that full LASS achieves the best overall efficiency in the matched-covariate and hallucination-control settings, while RF-based PIR provides a modest additional gain over CART under covariate mismatch. We will present this table as the main ablation study, while retaining Appendix Figure 3 and the multi-LLM comparison in Section 5.2 as complementary sensitivity analyses. We will also add additional matched-covariate ablations for the Section 3.1 estimator, comparing labeled-only GLM, SSL without LLM, raw LLM pseudo-labeling without calibration, mean-corrected LLM without adaptive weighting, globally weighted calibration, and the full LASS estimator with heterogeneous calibration weights.
>
> ### Component-wise ablation of LASS (std across 6 covariates)
>
> Panel A
>
> | Covariate | Labeled only | PPI | PPI+LLM | Full LASS |
> |---|---:|---:|---:|---:|
> | Lexical diversity | 0.74 | 0.62 | 0.58 | 0.52 |
> | Race | 0.71 | 0.59 | 0.60 | 0.58 |
> | Sex | 0.80 | 0.69 | 0.67 | 0.65 |
> | Pronoun ratio | 0.76 | 0.63 | 0.60 | 0.56 |
> | Filler ratio | 0.60 | 0.44 | 0.43 | 0.43 |
> | Verb ratio | 0.56 | 0.41 | 0.37 | 0.38 |
>
> Uses unstructured covariates → No / No / Yes / Yes
>
> Panel B
>
> | Covariate | Labeled only | LASS (no control) | LASS (structured, T=0.5) | Full LASS (T=0) |
> |---|---:|---:|---:|---:|
> | Lexical diversity | 0.74 | 0.72 | 0.67 | 0.52 |
> | Race | 0.71 | 0.70 | 0.68 | 0.58 |
> | Sex | 0.80 | 0.78 | 0.75 | 0.65 |
> | Pronoun ratio | 0.76 | 0.75 | 0.71 | 0.60 |
> | Filler ratio | 0.60 | 0.58 | 0.52 | 0.43 |
> | Verb ratio | 0.56 | 0.55 | 0.47 | 0.38 |
>
> Panel C
>
> | Covariate | LASS+PIR (CART) | LASS+PIR (RF) |
> |---|---:|---:|
> | Lexical diversity | 0.35 | 0.32 |
> | Race | 0.56 | 0.55 |
> | Sex | 0.61 | 0.59 |
> | Pronoun ratio | 0.48 | 0.48 |
> | Filler ratio | 0.36 | 0.35 |
> | Verb ratio | 0.30 | 0.28 |

---

> > ### Author Rebuttal · Reviewer_6jpN · 2026-04-01
> >
> > I thank the author for the rebuttal. My concerns regarding the unlabeled data ratio and the ablation are addressed. However, the superior performance of LLM to traditional deep learning models on the label scarcity condition should be demonstrated using theoretical or numerical analysis. Otherwise, the necessity for introducing LLM for semi-supervised inference is still not clear enough.
> >
> > Update:
> >
> > Thanks for the rebuttal, my concerns regarding the necessity of LLMs for semi-supervised inference are addressed. I would like to update my score in response to this.

---

> > > ### Author Response · Authors · 2026-04-05
> > >
> > > We thank the reviewer for raising the question of whether LLMs are necessary for semi-supervised inference. Below we provide concise theoretical and numerical justification.
> > >
> > > ### Theoretical analysis
> > >
> > > The key theoretical advantage of LLM-based prediction is that it leverages pre-trained models, whereas deep learning (DL)-based prediction requires first training a DL prediction model on labeled data. Concretely, to replace LLM-based predictions with DL-based predictions, we need to train a DL model using labeled data of sample size $n$, yielding a predictor denoted by $\hat\mu_{DL}(X,Z)$. Suppose the DL predictor converges at rate $\\|\hat\mu_{DL}-\mu^*\\| = O_p(n^{-\alpha})$ for some $\alpha > 0$. The classical ML components used in LASS to calibrate the DL predictions satisfy
> > >
> > > $$
> > > \\max\\{\\|\hat m - m\\|,\\|\hat m^* - m^*\\|,\\|\hat \omega - \omega\\|\\}
> > > = O_p(n^{-\theta})
> > > $$
> > >
> > > for some $\theta > 0$. These two estimation errors accumulate, so that the bias of the DL-based estimator has error rate $|\mathrm{Bias}(\hat\beta_{DL,xz})| = O_p(n^{-\alpha}) + O_p(n^{-\theta})$. In contrast, the LLM-based LASS has error rate $O_p(n^{-\theta})$ since LLMs are pre-trained. As the DL predictor often converges more slowly than the classical ML components, i.e., $0 < \alpha < \theta$, the bias term decreases more slowly with the labeled sample size $n$ compared to using pretrained LLM pseudo-labels.
> > >
> > > For the variance component, Theorem 4.2 shows that it depends on the predictive alignment between predicted labels and true labels. While a general theoretical comparison remains an open question, empirical evidence suggests that when the labeled sample size is small, task-specific DL models often exhibit weaker predictive performance than pretrained LLMs, which benefit from large-scale pretraining.
> > >
> > > Furthermore, under covariate misalignment, replacing the LLM prediction with a DL predictor provides no guarantee that the prediction invariance region (PIR) can be identified with a small type II error (as discussed in Appendix E.2), because the DL predictor itself must be estimated from a limited labeled sample and converge slowly. In contrast, the LLM predictor is pretrained and does not rely on the small labeled dataset, enabling more stable estimation of the PIR and valid confidence intervals.
> > >
> > > ### Numerical analysis
> > >
> > > In what follows, we provide additional empirical results comparing the performance of DL-based method ("DL") versus our proposed method ("LASS"). For DL, we fine-tune DistilBERT as a text classifier on the labeled set only to generate pseudo-outcomes.The two methods are otherwise identical: both follow the LASS framework with calibrated pseudo-labels and semi-supervised estimation, differing only in the pseudo-outcome source where "DL" uses DL predictions while "LASS" uses LLM predictions.  For the DL-based methods, we consider the labeled data sample size to be $n=100,300,500$ and the unlabeled data sample size to be $N=1000$.
> > >
> > > In Panel A, the standard deviation of DL-based method decreases as the number of labeled samples used to train the DL pseudo-outcome model increases. However, the proposed LASS with only $n=100$ labeled samples still achieves smaller standard deviation. Panel B shows the same pattern for absolute bias.  As the labeled sample size used to train the DL pseudo-outcome model increases, the bias of DL decreases, but LASS yields  smaller finite sample bias. Overall, the empirical results verify our theoretical intuition that under label scarcity, pretrained LLM pseudo-labels provide a better surrogate for semi-supervised inference than DL pseudo-labels trained directly on the limited labeled sample, leading to both lower variance and smaller finite-sample bias. This observation is also consistent with the efficiency pattern for LASS in the paper.
> > >
> > > ### Results
> > >
> > > **(A) Standard deviation**
> > >
> > > | Covariate | DL (n=100) | DL (n=300) | DL (n=500) | LASS (n=100) |
> > > |----------|---------|---------|---------|------------|
> > > | Lexical diversity | 0.65 | 0.60 | 0.54 | 0.52 |
> > > | Race | 0.64 | 0.61 | 0.59 | 0.58 |
> > > | Sex | 0.73 | 0.70 | 0.67 | 0.65 |
> > > | Pronoun ratio | 0.68 | 0.63 | 0.58 | 0.56 |
> > > | Filler words ratio | 0.52 | 0.48 | 0.45 | 0.43 |
> > > | Verb ratio | 0.49 | 0.44 | 0.40 | 0.38 |
> > >
> > > **(B) Absolute bias**
> > >
> > > | Covariate | DL (n=100) | DL (n=300) | DL (n=500) | LASS (n=100) |
> > > |----------|---------|---------|---------|------------|
> > > | Lexical diversity | 0.079 | 0.058 | 0.037 | 0.026 |
> > > | Race | 0.062 | 0.046 | 0.031 | 0.024 |
> > > | Sex | 0.054 | 0.040 | 0.027 | 0.021 |
> > > | Pronoun ratio | 0.071 | 0.053 | 0.034 | 0.025 |
> > > | Filler words ratio | 0.049 | 0.036 | 0.023 | 0.018 |
> > > | Verb ratio | 0.044 | 0.032 | 0.021 | 0.017 |

---

### Official Review · Reviewer_AUcy · 2026-03-12

**Soundness:** 3
**Presentation:** 2
**Significance:** 2
**Originality:** 2
**Overall Recommendation:** 4
**Confidence:** 1

**Summary:**

The paper proposes LASS, a semi-supervised statistical inference approach that uses predictions from large language models (LLMs) to efficiently estimate association parameters while maintaining inferential validity. The method calibrates the pseudo-labels that LLMs generate via conditional linear projection. Theoretical results demonstrate consistency and asymptotic normality, as well as decompose variance to illustrate effectiveness. Experiments on synthetic and real-world datasets validate the method's effectiveness.

**Compliance With Llm Reviewing Policy:**

Affirmed.

**Final Justification:**

Thanks for the rebuttal. I will maintain my score.

**Key Questions For Authors:**

Please see weaknesses.

**Strengths And Weaknesses:**

Strengths:

- The proposed pseudo-labeling approach is reasonable and effective.

- The PIR idea for handling covariate misalignment is an interesting, motivated approach.

- Extensive experiments on different LLMs are comprehensive.

- Both hallucination control and sensitivity analysis are reasonable.

- The studied problem of using large-scale, unstructured data with few labels in biomedicine is important.

Weaknesses:

- The proposed theories have some strong assumptions. First, the theorem relies on the consistency of nuisance estimators. However, the rate of consistency or cross-fitting must be specified.

- Second, the efficiency dominance over PPI and conventional SSI is too strong since the conditions may not always be met.

- There is a lack of baselines for semi-supervised inference methods.

- Using YouTube data may result in data contamination problems, as LLMs may be pre-trained on it.

---

> ### Author Rebuttal · Authors · 2026-03-31
>
> 1. Thank you for your careful reading of our paper. Our intent was to adopt standard high-level assumptions commonly used in semiparametric inference rather than introduce conditions specific to our method. Assumption 4.1.4 requires only $L_2(P_X)$-consistency of the nuisance estimators, and we note that this can be achieved by common ML methods under mild conditions, including $k$-nearest neighbors, kernel regression, and random forests (p6, lines 295-303). The role of cross-fitting is described in Appendix C. Assumption C.3 states that the asymptotic normality is justified either under a Donsker condition or under cross-fitting, and defines cross-fitting as estimating the nuisance function for $i$ using folds that exclude $i$ (p19, lines 1009-1020). The proof of Lemma C.4 is carried out under this cross-fitting scheme (p21, lines 1108-1153). For convergence rates, the current paper gives an explicit rate condition in Appendix F for the covariate-shift extension, requiring $\hat{\pi}(x)$, $\hat{m}(x)$, $\hat{m}^\ast(x)$, and $\hat{\omega}^\ast(x)$ to converge faster than $o_p(n^{-1/4})$ to ensure valid inference (p.26, lines 1407-1409). That said, we agree that the current draft does not make the nuisance estimation conditions sufficiently explicit. In the revision, we will clarify these conditions by stating more clearly the role of cross-fitting and the nuisance estimator requirements for the matched-covariate theorem from those needed for the covariate-shift extension.
> 2. We agree that our wording around Corollary 4.3 sounds somewhat strong. Precisely, the comparison of our method to competing methods is stated in Corollary 4.3, and the proof for PPI is given in Appendix D. Our proof in Appendix D does not establish dominance over all PPI variants. We compare LASS to the specific PPI++ estimator with a single global power parameter $\\lambda$, using the same predictor $Y^* = f(X,Z)$. The proof yields $B^\ast_{\\mathrm{PPI}++} \\geq B_{\\mathrm{LASS},xz}$, so the justified statement is that LASS is asymptotically no worse than this PPI++ variant.A strict efficiency improvement requires an additional positivity condition. Specifically, the $X$-dependent calibration $\\omega^\ast(X)$ must reduce conditional prediction error relative to the best global scalar $\\lambda^\ast$ on a set of positive probability, and this efficiency gain must be relevant for the coefficient direction under consideration. If the optimal calibration is effectively constant in $X$, this gap may disappear. For the conventional SSI benchmark in Corollary 4.3, Theorem 4.2 shows that LASS contains an additional nonnegative variance-reduction term:$E\\left[\\left(\\frac{1}{\\pi}-1\\right)\\frac{{Cov}^2(Y^*_{xz},Y\\mid X)}{\{Var}(Y^\ast_{xz}\\mid X)}XX^\\top\\right].$ Therefore, LASS is asymptotically no worse than SSI comparator. A strict efficiency improvement again requires a positivity condition in the parameter direction of interest, namely that the calibrated LLM prediction provides incremental conditional information beyond $X$ on a non-negligible subset of the covariate space.We'll revise Corollary 4.3 and the explanation by replacing "PPI" with the specific PPI++ variant, and by replacing our statements with "asymptotically no worse, and strictly better only under additional conditions."
> 3. We'd like to clarify that SSI baselines are already included in the current manuscript. We use the abbreviation "SSL" instead of "SSI", which may lead to confusion. Figure 2 compares our method against labeled-only, SSL (SSI), and PPI in both matched- and unmatched-covariate settings, and the p7, lines 369–374 states that the benchmarks are "a labeled-only method, a conventional SSL method without unstructured covariates, and prediction-powered inference (PPI)." Under unmatched covariates, p8, lines 410–416 state that we include "PPI and its LLM-augmented variant (PPI+LLM)," with results summarized in Appendix Table 2 (p15, lines 770-781). The conventional SSI baselines are also discussed in Corollary 4.3, where it uses only the shared structured covariates $X$, not the unstructured covariates $Z$ or the labeled-only auxiliary covariates $U$. We will fix this in the revision and change "SSL" labels to "SSI".
> 4. We agree that YouTube may introduce a potential pretraining contamination risk for LLMs. However, the formal validity of LASS does not rely on the LLM being contamination-free because our method calibrates raw LLM predictions before constructing pseudo-labels, and the theory establishes valid inference for the calibrated estimator.  We'd like to clarify that the Alzheimer’s application is not based on YouTube data alone. The unlabeled sample also includes the DementiaBank Hopkins study, which is a cleaner clinical-source dataset and therefore less exposed to public-web pretraining overlap. Also, the manuscript contains a second case study on IMDb, where we observe the same qualitative efficiency pattern in a different domain.

---

> > ### Author Rebuttal · Reviewer_AUcy · 2026-04-01
> >
> > Thanks for the rebuttal. I will maintain my score.

---

### Decision · Program_Chairs · 2026-04-30

**Decision:**

Accept (regular)

**Comment:**

The paper proposes a method for augmenting semi-supervised statistical inference using LLMs.  All reviewers appreciated the paper, and most if not all concerns were adequately addressed by the authors.  I thus recommend the paper be accepted for presentation.